# Directed transport of neutrophil-derived extracellular vesicles enables platelet-mediated innate immune response

Jan Rossaint[1], Katharina Kühne[1], Jennifer Skupski[1], Hugo Van Aken[1], Mark R. Looney[2], Andres Hidalgo[3,4] & Alexander Zarbock[1]

The innate immune response to bacterial infections requires the interaction of neutrophils and platelets. Here, we show that a multistep reciprocal crosstalk exists between these two cell types, ultimately facilitating neutrophil influx into the lung to eliminate infections. Activated platelets adhere to intravascular neutrophils through P-selectin/P-selectin glycoprotein ligand-1 (PSGL-1)-mediated binding, a primary interaction that allows platelets glycoprotein Ibα (GPIbα)-induced generation of neutrophil-derived extracellular vesicles (EV). EV production is directed by exocytosis and allows shuttling of arachidonic acid into platelets. EVs are then specifically internalized into platelets in a Mac1-dependent fashion, and relocated into intracellular compartments enriched in cyclooxygenase1 (Cox1), an enzyme processing arachidonic acid to synthesize thromboxane $A_2$ ($TxA_2$). Finally, platelet-derived-$TxA_2$ elicits a full neutrophil response by inducing the endothelial expression of ICAM-1, intravascular crawling, and extravasation. We conclude that critical substrate–enzyme pairs are compartmentalized in neutrophils and platelets during steady state limiting non-specific inflammation, but bacterial infection triggers regulated EV shuttling resulting in robust inflammation and pathogen clearance.

[1] Department of Anaesthesiology, Intensive Care and Pain Medicine, University Hospital Münster, 48149 Münster, Germany. [2] Department of Medicine, University of California, San Francisco, California 94143, USA. [3] Institute for Cardiovascular Prevention, Ludwig-Maximilians-University, 80336 Munich, Germany. [4] Area of Cell and Developmental Biology, CNIC, 28029 Madrid, Spain. Correspondence and requests for materials should be addressed to A.Z. (email: zarbock@uni-muenster.de).

The acute respiratory distress syndrome (ARDS) is a life threatening disease with a high incidence[1]. Despite improved supportive care, the mortality of ARDS remains high at ∼40% (ref. 1). ARDS is characterized by an increased number of neutrophils in the lung and increased permeability, leading to lung oedema and consequently to decreased pulmonary gas exchange[1,2]. Major causes for the development of ARDS are pneumonia and sepsis, and Gram-negative bacteria are the dominant pathogens[3]. The recruitment of neutrophils into inflamed tissue is required for eliminating invading pathogens, but they are also involved in tissue destruction by releasing a variety of enzymes[4]. Extravasation of neutrophils in peripheral tissues proceeds in a cascade-like fashion[4], whereas the mechanisms of neutrophil recruitment into the inflamed lung are still poorly defined[5]. During pneumonia, neutrophils may also form heterotypic aggregates with other blood-born cells such as platelets[6]. This interaction between platelets and neutrophils promotes neutrophil recruitment and activation[7–11], thus modulating the innate immune response[12]. Recent studies provide evidence for the significance of platelets in mouse models of acid-induced acute lung injury[9,13] and transfusion-related acute lung injury[7,10,11,14]. During inflammation, platelets accumulate at sites of inflamed vascular endothelium and present P-selectin on their surface. P-selectin can bind to PSGL-1 on circulating neutrophils, which then adhere to platelets[6,15,16]. Apart from P-selectin binding to PSGL-1, bonds between activated platelets and neutrophils may also be formed by the platelet integrin $\alpha_{IIb}\beta_{III}$ (GPIIbIIIa) binding to the neutrophil integrin $\alpha_M\beta_2$ (Mac-1) via fibrinogen as well as direct binding of Mac-1 on neutrophils to platelet GPIb$\alpha$ (refs 6,17,18). The interaction of platelets with neutrophils fully activates neutrophils[8,10,19]. During inflammatory processes neutrophils may generate extracellular vesicles (EV)[20]. EVs are actively secreted from neutrophils and may contain certain subsets of cytosolic and membrane-bound molecules. Previous reports suggest that the generation and liberation of EVs derived from various cells is a highly organized process involving cell-autonomous excretory mechanisms and suggests that the uptake of EVs into target cells is also mediated by distinct molecular mechanisms[21,22]. However, the exact role of EVs in inflammation, particularly in platelet–neutrophil interactions, and the molecular mechanism regulating their excretion and uptake remain poorly defined.

The interaction of platelets and neutrophils leads to neutrophil activation by integrin-mediated outside-in-signalling in addition to the presentation of chemokines and lipid mediators by platelets to neutrophils[23–27]. One important lipid mediator is thromboxane A2 ($TxA_2$)[28]. $TxA_2$ is an arachidonic acid metabolite which is generated in several cell types and tissues, such as platelets, inflammatory cells and pulmonary tissue, by the enzymes cyclooxygenase, hydroperoxidase and tissue-specific isomerases[29]. The biosynthetic pathway for $TxA_2$ production is mostly shared with that of other prostaglandins. Phospholipase A2 releases arachidonic acid from membrane phospholipids, which is the substrate for a metabolic pathway involving cyclooxygenase (Cox)-1 and Cox2 and hydroperoxidase to form prostaglandin H2 ($PGH_2$). Thromboxane synthase is the most abundant isomerase in platelets and converts $PGH_2$ to $TxA_2$, which in turn has a short half-life and is converted to its stable metabolite thromboxane B2 ($TxB_2$).

The aim of the present study was to investigate the molecular mechanisms by which neutrophils contribute to thromboxane generation in platelets and the pathophysiological implications of this process during bacterial infection. Unexpectedly, we found that this process is enabled through cooperative metabolic processing and metabolite transport via EVs between neutrophils and platelets.

## Results

**$TxA_2$ generation is regulated by complex formation**. Cox1 is involved in the production of prostaglandins, such as $TxA_2$, from arachidonic acid. We analysed thromboxane production from platelets and neutrophils individually, and in combination, after stimulation with ADP and fMLP, two potent agonists for platelets and neutrophils, respectively. Production of $TxB_2$, the stable metabolite of $TxA_2$, was significantly increased when platelets and neutrophils were co-incubated, and blocking the physical interaction between both cell types by pretreating the cells with blocking antibodies against either P-selectin or PSGL-1 decreased $TxB_2$ production (Fig. 1a). Because the two cells can synthesize prostaglandins, we next investigated the cellular source of arachidonic acid required for $TxB_2$ production. We incubated neutrophils, platelets or both cell types with radioactively labelled arachidonic acid ($C^{14}$-AA) and measured the production of radio-labelled thromboxane B2 ($TxB_2$-$C^{14}$) after stimulation of both cell types with ADP and fMLP. This assay revealed that neutrophils provide a significant amount of arachidonic acid and that maximum $TxB_2$ production is only achieved when both cellular sourced of arachidonic acid (that is, neutrophils and platelets) cooperate in prostaglandin production (Fig. 1b). To ensure equal labelling of platelets and neutrophils in this assay, the radioactive counts per minute (c.p.m.) of cell lysates with equal protein content (indicating equal cell masses) were analysed using a $\beta$-counter to verify equal $C^{14}$-AA loading. To differentiate the role of Cox1 and Cox2 in platelets and neutrophils in $TxB_2$-$C^{14}$ production using arachidonic acid from neutrophils, we isolated neutrophils and platelets from wild-type (WT), $Cox1^{-/-}$ and $Cox2^{-/-}$ mice and incubated only the neutrophils with AA-$C^{14}$. Afterwards, neutrophils and platelets were coincubated and $TxB_2$-$C^{14}$ production and total $TxB_2$ levels were analysed. We observed that only Cox1 in platelets is necessary for $TxB_2$ production utilizing arachidonic acid from neutrophils (Fig. 1c,d). To identify molecular adhesion molecules required for intercellular interactions and exchange of arachidonic acid during thromboxane production by platelets, we used a blocking anti-P-selectin antibody (RB40.34) and anti-PSGL-1 antibody (4RA10) as well as tirofiban (antagonist of the platelet integrin $\alpha_{IIb}\beta_{III}$) and demonstrated that $TxB_2$-$C^{14}$ production and overall $TxB_2$ production was reduced after blockade of P-selectin or PSGL-1, but not after inhibiting the platelet integrin $\alpha_{IIb}\beta_{III}$ (Fig. 1e,f). Notably, the activity of Cox1 was increased in the presence of neutrophils, and again this effect was reversed after blocking P-selectin or PSGL-1, but not $\alpha_{IIb}\beta_{III}$ (Fig. 1g).

***Escherichia coli*-induced pneumonia is platelet dependent**. Platelet depletion has been shown to reduce neutrophil recruitment, vascular permeability, and tissue destruction in sterile models of acute lung inflammation[7,9,10,13]. To investigate whether platelets potentiate productive neutrophil recruitment and bacterial clearance during pneumonia triggered by *E. coli*, mice were platelet depleted by triple injections of busulfan as reported previously[8]. Intratracheal instillation of *E. coli* in platelet-depleted mice led to a significantly reduced survival rate compared with control mice (Fig. 2a). *E. coli* instillation produced an increased neutrophil count in the BAL (Fig. 2b) and a detectable number of colony-forming units (c.f.u's) in the bronchoalveolar lavage (BAL), lung and spleen (Fig. 2c–e). Platelet depletion by busulfan decreased the number of neutrophils in the BAL (Fig. 2b) and significantly increased bacterial burden in the BAL, lung and spleen (Fig. 2c–e) after *E. coli* instillation compared with vehicle-treated mice. The same effect was observed when platelets were depleted with a different approach utilizing a platelet-depleting antibody

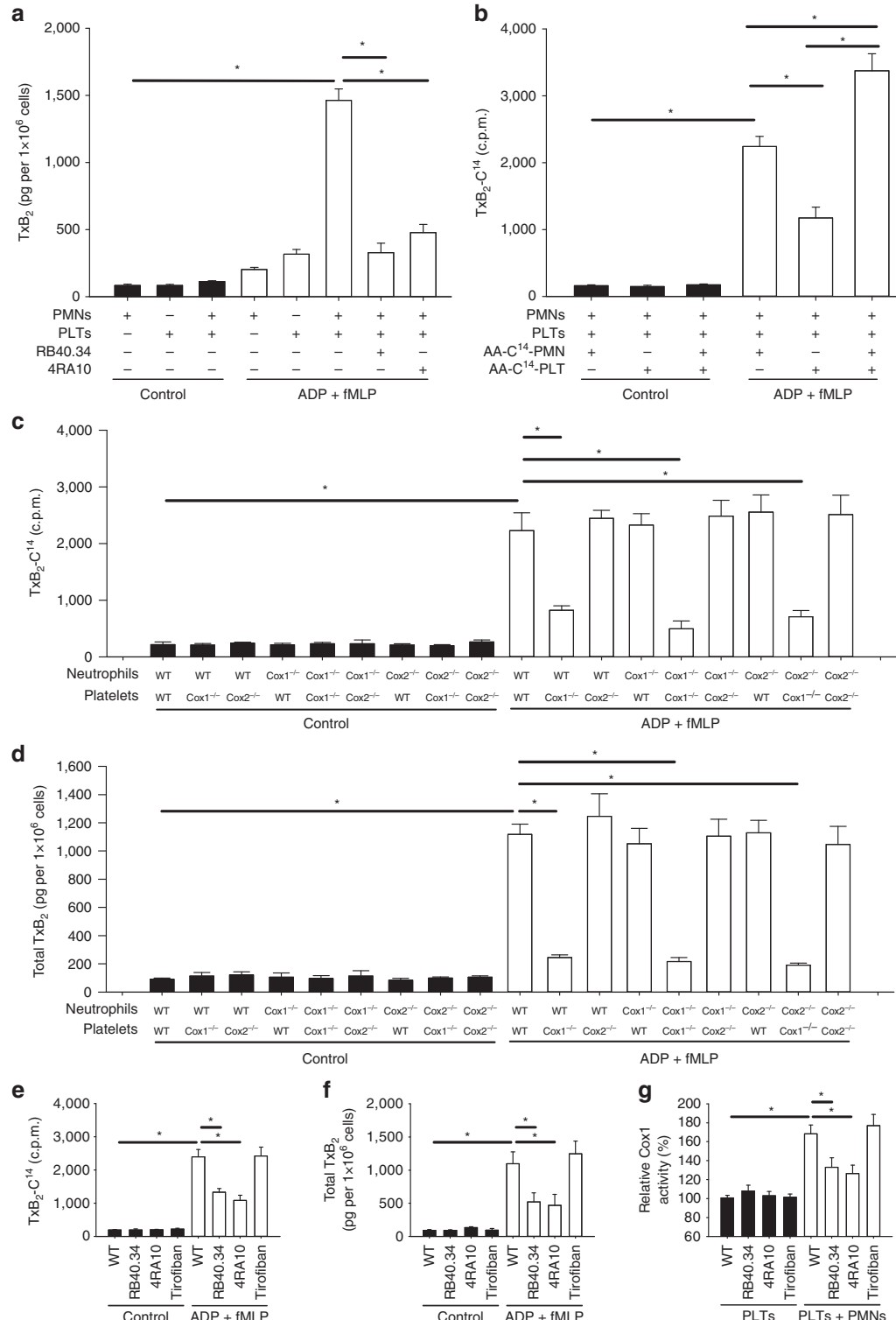

**Figure 1 | Thromboxane generation by platelets is enabled by interacting with neutrophils.** (**a**) PMNs and platelets were isolated from WT mice and $TxB_2$ production in both cell types alone as well as after co-incubation under control conditions or stimulation with ADP (10 µM) and fMLP (10 µM) was analysed in the presence or absence of blocking antibodies against P-selectin (clone RB40.34. 5 µg ml$^{-1}$) or PSGL-1 (4RA10, 5 µg ml$^{-1}$) ($n = 3$). (**b**) PMNs and/or platelets from WT mice were radioactively labelled with $C^{14}$-AA and $TxB_2$-$C^{14}$ in control and ADP/fMLP-stimulated samples was measured ($n = 3$). PMNs and platelets were isolated from WT mice, $Cox1^{-/-}$ and $Cox2^{-/-}$ mice and only PMN were radioactively labelled with $C^{14}$-AA. (**c**) $TxB_2$-$C^{14}$ in control and ADP/fMLP-stimulated samples ($n = 3$). (**d**) Total $TxB_2$ production in control and ADP/fMLP-stimulated samples ($n = 3$). PMNs and platelets were isolated from WT mice and treated with blocking antibodies against P-selectin (clone RB40.34, 5 µg ml$^{-1}$), PSGL-1 (clone 4RA10, 5 µg ml$^{-1}$) or tirofiban (100 µM). (**e**) $TxB_2$-$C^{14}$ in control and ADP/fMLP-stimulated samples ($n = 3$). (**f**) Total $TxB_2$ production in control and ADP/fMLP-stimulated samples ($n = 3$). (**g**) Cox1 activity in platelets alone and after co-incubation with neutrophils ($n = 3$). Mean ± s.e.m., ANOVA plus Bonferroni testing, *$P < 0.05$.

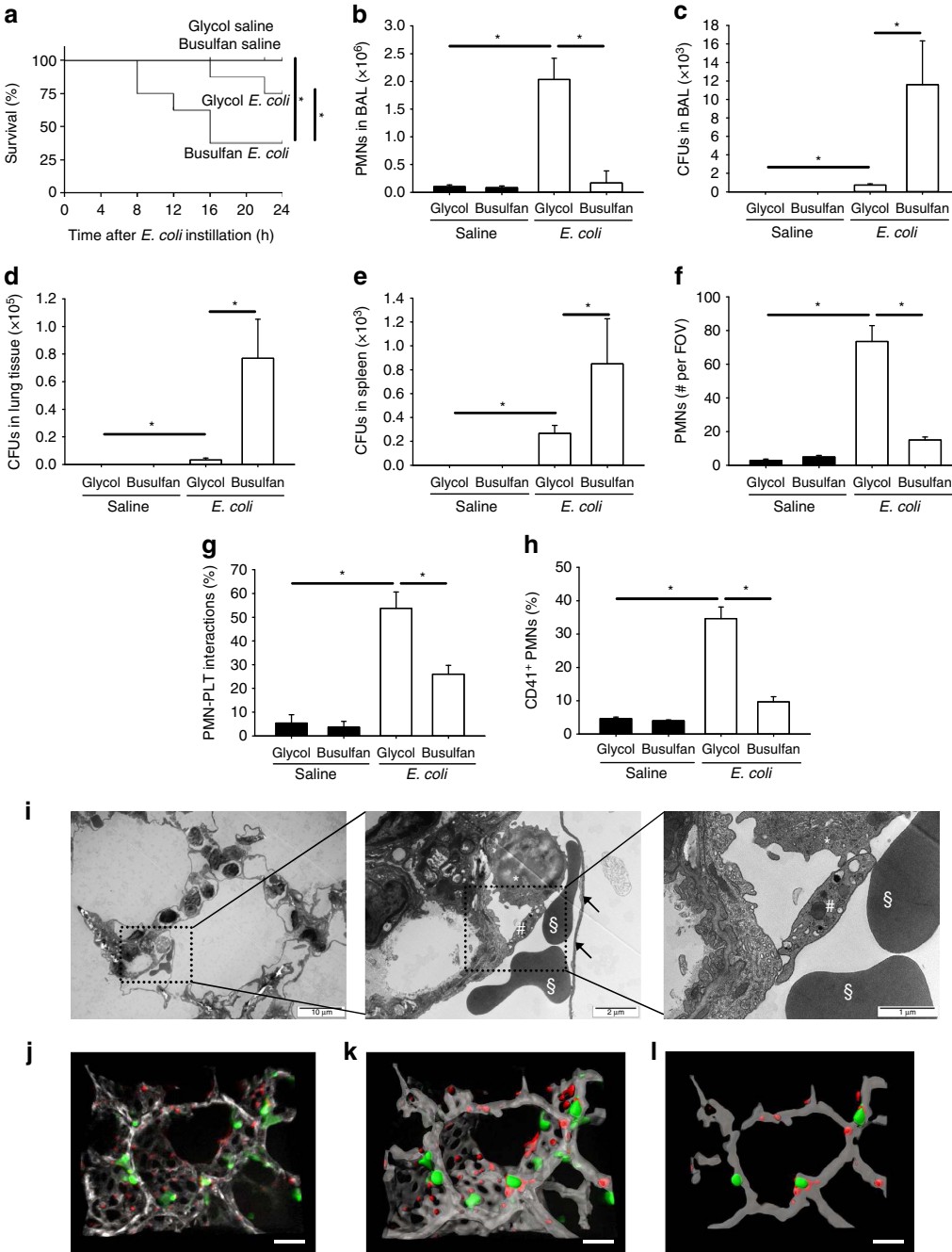

**Figure 2 | Platelets are required host defence during *E. coli*-induced pneumonia.** Glycol- and busulfan-treated wild-type mice were injected intratracheally with saline or viable *E. coli*. (**a**) Survival 24 h after instillation of $8 \times 10^6$ viable *E. coli* ($n = 11$–15). (**b**) Neutrophil recruitment into the alveoli and the c.f.u. count in the BAL (**c**), lung tissue (**d**) and the spleen (**e**) were analysed 24 h after intratracheal instillation of $6 \times 10^6$ c.f.u.'s per mouse ($n = 4$). Neutrophil accumulation in the lung was visualized by intravital microscopy of the middle right lung lobe by intravital microscopy. (**f**) Number of accumulated neutrophils per field of view (FOV) ($n = 3$). (**g**) Neutrophils interacting with platelets in the lung capillaries *in vivo* ($n = 3$). (**h**) The formation of circulating platelet–neutrophil aggregates in the blood of glycol- and busulfan-treated mice after intratracheal instillation of saline or viable E. coli ($6 \times 10^6$ c.f.u.'s per mouse) was measured by flow cytometry ($n = 4$). (**i**) Ultrathin cross-sectioned lung tissue imaged by transmission electron microscopy from lung tissue of WT mice after inducing pneumonia showing a neutrophil (*) in close proximity to a platelet (#) and 2 erythrocytes (§) within the boundaries of the capillary wall (black arrow). (**j**) Confocal image of lung tissue from WT mice after induction of *E. coli* pneumonia with (**k**) 3D reconstruction and (**l**) exemplary display of a single confocal plane to identify neutrophils (Ly6G, green) and platelets (CD41, red) within pulmonary capillaries stained with PECAM-1 antibody (gray) (scale bars equal 20 μm). Mean ± s.e.m., ANOVA plus Bonferroni testing, $*P < 0.05$.

(Supplementary Fig. 2a–d). In a murine model of pneumonia induced by intratracheal instillation of the clinically relevant pathogen *Klebsiella pneumoniae*, platelet depletion also decreased the number of neutrophils in the BAL (Supplementary Fig. 1a) and significantly increased bacterial burden in the BAL, lung and

spleen (Supplementary Fig. 1b–d). To exclude adverse effects of busulfan treatment on neutrophil function, we isolated neutrophils from control- and busulfan-treated mice and analysed ICAM-1 binding, transmigration and phagocytosis of *E. coli* particles *in vitro*. These experiments showed that all these neutrophil

functions were not significantly altered by busulfan (Supplementary Fig. 2e–g). To directly visualize neutrophil accumulation in the lung during *E. coli*-induced pneumonia, we employed intravital microscopy in living mice. Instillation of *E. coli* caused a significant increase in neutrophil accumulation in the lung microvasculature, which was markedly reduced in platelet-depleted mice (Fig. 2f). In addition, platelet depletion caused an expected reduction in the number of platelet–neutrophil aggregates (Fig. 2g). To investigate circulating platelet–neutrophil aggregates, we used a flow cytometry based analysis method[9]. The number of circulating platelet–neutrophil aggregates in the peripheral blood significantly increased after instillation of *E. coli* (Fig. 2h), whereas platelet depletion significantly decreased the amount of circulating platelet–neutrophil aggregates (Fig. 2h). The presence of platelet–neutrophil aggregates in the microcirculation of the lung after *E. coli* instillation was also demonstrated by transmission electron microscopy (Fig. 2i). To further visualize the allocation of platelets and neutrophils during *E. coli* induced pneumonia we stained neutrophils, platelets and PECAM as an endothelial marker of the pulmonary microvasculature and analysed lung sections from *E. coli*-treated animals by confocal microscopy (Fig. 2j–l). Together, these data demonstrate that platelets activate neutrophil recruitment during bacterial infection.

**Host defence from pneumonia requires hematopoietic Cox1.** TxA$_2$ has been involved in the pathogenesis of pneumonia[9]. To investigate whether Cox1 or Cox2 are necessary for neutrophil recruitment and bacterial clearance during *E. coli*-induced pneumonia, we intratracheally instilled viable *E. coli* into WT, Cox1$^{-/-}$ and Cox2$^{-/-}$ mice. Cox1 deficiency, but not Cox2 deficiency, caused a significantly decreased number of recruited neutrophils in the BAL after *E. coli* instillation (Supplementary Fig. 3a). Consequently, Cox1$^{-/-}$ mice had a significantly higher c.f.u. count in the BAL, lung and spleen (Supplementary Fig. 3b–d) following *E. coli* instillation compared with WT and Cox2$^{-/-}$ mice. By using bone marrow chimeric mice, we demonstrated that Cox1 deficiency in hematopoietic cells caused a reduced number of neutrophils in the BAL and impaired bacterial clearance, as indicated by increased c.f.u.'s in the BAL, lung and spleen (Supplementary Fig. 3e–h). Instillation of *E. coli* caused a significant higher number of neutrophils in the lung microvasculature of WT mice compared with Cox1$^{-/-}$ mice, as revealed by intravital microscopy (Supplementary Figs 3i and 4). In addition, platelet–neutrophil interactions were significantly decreased in Cox1$^{-/-}$ mice compared with WT mice after *E. coli* instillation (Supplementary Figs. 3j and 4). Cox1 deficiency led to a significantly reduced survival rate compared with vehicle-treated control mice (Supplementary Fig. 3k). Likewise, Cox1-deficient mice after infection with *K. pneumoniae* also had an impaired neutrophil recruitment into the lung (Supplementary Fig. 1a) resulting in higher c.f.u. counts in the BAL, lung tissue and spleen compared with control animals (Supplementary Fig. 1b–d). These findings are consistent with our previous observation that platelet-derived Cox1, but not Cox2, is needed for the synthesis of TxA$_2$ upon contact with neutrophils (Fig. 1).

**Blocking the thromboxane receptor aggravates pneumonia.** Our data indicate that Cox1 in platelets is required for thromboxane A$_2$ production, neutrophil recruitment, and bacterial clearance during *E. coli* and *K. pneumoniae*-induced pneumonia. To demonstrate that the interaction of platelets and neutrophils is necessary for adequate thromboxane production *in vivo*, we measured the serum TxB$_2$ levels in mice after inducing *E. coli*-induced pneumonia and found that serum TxB$_2$ levels are significantly decreased in platelet-depleted mice after *E. coli*

instillation (Fig. 3a). Neutrophil depletion, or blockade of P-selectin or PSGL-1 significantly decreased serum TxB$_2$ levels in mice after inducing *E. coli*-induced pneumonia (Fig. 3a). Thus neutrophil–platelet contacts mediated by PSGL-1 is required for thromboxane production during pulmonary infection. To investigate the role of the thromboxane receptor during *E. coli*-induced pneumonia, we pretreated mice with a thromboxane receptor inhibitor (SQ 29548) and found a significantly decreased number of neutrophils in the BAL (Fig. 3b) and a significantly increased number of c.f.u.'s in the BAL, lung and spleen after *E. coli* instillation (Fig. 3c–e). Blocking P-selectin or PSGL-1 also reduced the number of neutrophils in the BAL (Fig. 3b) and increased the number of c.f.u.'s in the BAL, lung and spleen (Fig. 3c–e). We previously reported impaired neutrophil recruitment to the lungs after blocking the platelet integrin α$_{IIb}$β$_{III}$ using tirofiban in a murine model of ventilator-associated lung injury, and we could also observe reduced neutrophil recruitment and impaired bacterial clearance in the *E. coli*-induced pneumonia model after tirofiban administration (Fig. 3b–e)[8]. To specifically demonstrate that platelet and not endothelial P-selectin is responsible for these effect we transplanted bone marrow from P-selectin-deficient donor mice (Selp$^{-/-}$) into lethally irradiated WT recipient mice and vice versa. Pneumonia was induced in these mice 6 weeks after bone marrow transplantation. Mice transplanted with Selp$^{-/-}$ bone marrow showed a significantly reduced number of neutrophils in the BAL (Supplementary Fig. 5a) and increased number of c.f.u.'s in the BAL, lung and spleen after *E. coli* instillation (Supplementary Fig. 5b–d) compared with control mice transplanted with WT bone marrow. Bacterial infections are known to cause a local and systemic inflammatory response including the release of formylated peptides and pro-inflammatory mediators causing platelet activation *in vivo*. To directly demonstrate upregulation of P-selectin surface expression on platelets *in vivo*, we induced pneumonia by *E. coli* instillation in WT mice, isolated circulating platelets from blood after 4, 12 and 24 h and analysed platelet P-selectin expression by flow cytometry. We demonstrated that platelets from *E. coli*-treated mice expressed significantly more P-selectin on their cell surface compared with platelets isolated from saline-treated mice (Supplementary Fig. 5e). Intravital microscopy revealed a reduced number of adherent neutrophils in the lung microvasculature in mice pretreated with the thromboxane receptor inhibitor after *E. coli* instillation (Fig. 3f). Consequently, thromboxane receptor blockade resulted in a reduced survival after inducing *E. coli*-induced pneumonia (Fig. 3g), which indicated that TxA$_2$ is required for neutrophil recruitment and host defence during *E. coli*-induced pneumonia.

**Neutrophil-derived EV transfer arachidonic acid to platelets.** Activated neutrophils may generate and release EVs under inflammatory conditions. The number of EVs released after ADP/fMLP (10 μM each) stimulation was strongly increased in the presence of platelets (Fig. 4a). This effect was inhibited by pretreatment of platelets with blocking antibodies against either P-selectin (clone RB40.34, 5 μg ml$^{-1}$) or GPIbα (clone Xia.B2, 5 μg ml$^{-1}$) (Fig. 4a). To investigate the presence of arachidonic acid in EVs, we isolated EVs from the supernatants of neutrophils co-cultured or not with platelets and assessed the levels of arachidonic acid. The incubation of neutrophils together with platelets, versus neutrophils alone, increased the amount of arachidonic acid in the EV fraction (Fig. 4b). Confocal imaging revealed that platelets readily take-up isolated and fluorescently-labelled neutrophil-derived EVs (Fig. 4c). Likewise, confocal imaging also showed that neutrophil-derived EVs were

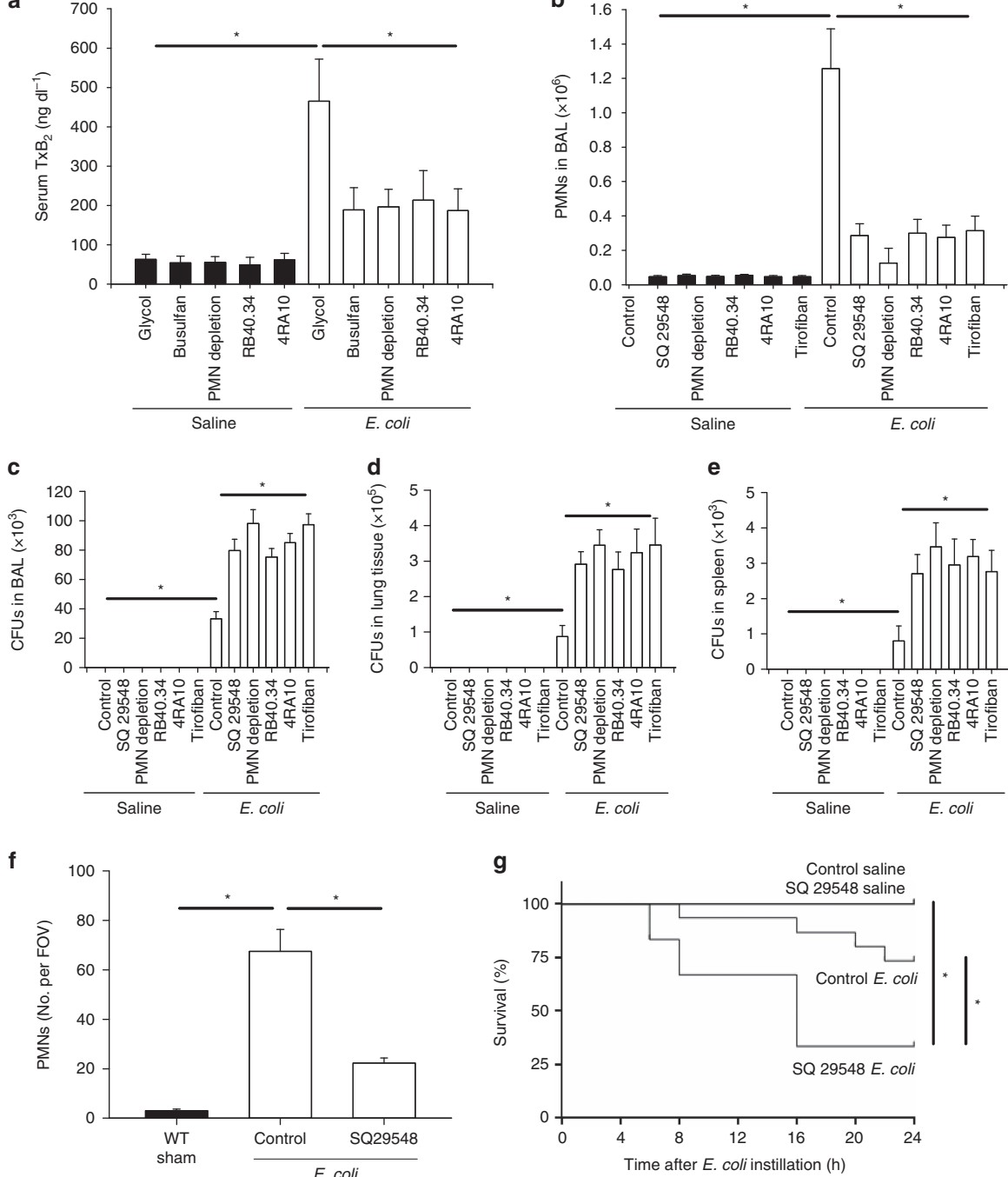

**Figure 3 | Blocking thromboxane receptors aggravates *E. coli*-induced pneumonia.** WT control mice, mice after injection of a thromboxane receptor antagonist (SQ 29548), following PMN depletion or after administration of blocking antibodies against P-selectin (clone RB40.34, 50 µg per mouse) or PSLG-1 (clone 4RA10, 50 µg per mouse), were instilled i.t. with viable *E. coli* ($6 \times 10^6$ c.f.u.'s per mouse) or saline. (**a**) Serum $TxB_2$ levels in glycol- and busulfan-treated mice, after PMN depletion and after blocking P-selectin (clone RB40.34, 50 µg per mouse) or PSLG-1 (clone 4RA10, 50 µg per mouse) ($n = 4$). (**b**) Neutrophil recruitment into the alveoli and the c.f.u. count in the BAL (**c**), lung tissue (**d**) and the spleen (**e**) were analysed after 6 h ($n = 4$). (**f**) The number of accumulated neutrophils in the lung was visualized by intravital microscopy ($n = 3$). (**g**) Survival 24 h after instillation of $8 \times 10^6$ viable *E. coli* ($n = 6$–15). Mean ± s.e.m., ANOVA plus Bonferroni testing, log rank test in 3 g *$P < 0.05$.

incorporated into platelets during platelet–neutrophil interactions (Fig. 4d) and, interestingly, neutrophil-derived EVs co-localized with platelet Cox1 (Fig. 4e). Having shown that the blocking GPIbα reduces EV release from neutrophils, we investigated thromboxane production and observed that GPIbα blockade reduces both the $TxB_2$-$C^{14}$ release, indicating reduced usage of neutrophil-derived arachidonic acid by platelets, as well as total

thromboxane production (Fig. 4f,g). In agreement with this finding, GPIbα blockade (clone Xia.B2, 5 µg ml$^{-1}$) reduced platelet Cox1 activity (Fig. 4h). The addition of isolated EVs to the platelet–neutrophil co-culture after GPIbα blockade was able to rescue $TxB_2$-$C^{14}$ release, total thromboxane production and platelet Cox1 activity (Fig. 4f–h). Consequently, blocking GPIbα caused a significantly decreased number of neutrophils in

 

the BAL after instillation of viable $6 \times 10^6$ c.f.u.'s of *E. coli*/mouse (Fig. 4j) and a significantly increase in the c.f.u. count in the BAL, lung and spleen (Fig. 4k–m) and the amount of circulating platelet–neutrophil aggregates in the blood stream (Fig. 4i). Importantly, reconstitution with neutrophil-derived EVs

$(2 \times 10^7$ per ml) improved survival in mice after blocking of GPIbα (Fig. 4n). Besides thromboxane, the biosynthesis of leukotrienes is also known to involve platelet–neutrophil interaction. While LTB4 synthesis by neutrophils relies on transcellular substrate transport from platelets to neutrophils, platelets require

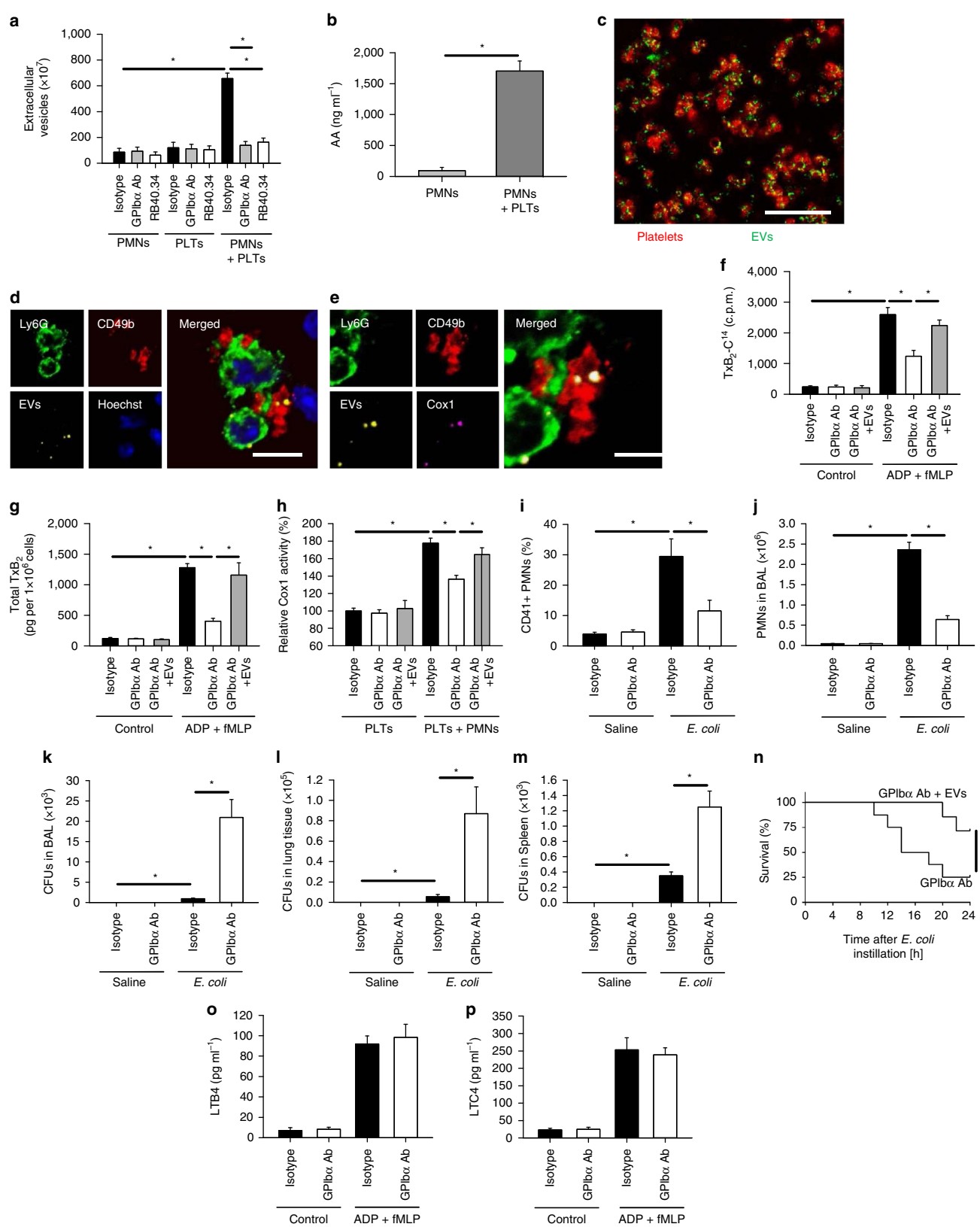

 

neutrophil-derived LTA4 for transformation to LTC4 (ref. 30). The concentrations of both LTB4 and LTC4 were increased under stimulated compared with control conditions. However, blocking GPIbα did not significantly impact LTB4 or LTC4 production, indicating the implication of different modulatory molecular mechanisms in comparison to thromboxane biosynthesis (Fig. 4o,p). Together, these findings indicated that $TxA_2$ production and host defence is dependent on platelet-borne Cox1, which utilizes neutrophil-derived arachidonic acid shuttled into platelets via EVs.

**EV shuttling involves directed release and uptake mechanisms.** The spatiotemporal proximity of the neutrophil–platelet interactions implies that the generation of neutrophil-derived EVs may be a directed process. Caveolin-1 and clathrin are known to be involved into cellular vesicle transport by regulating endosomal sorting and exocytosis, and vesicle internalization, respectively[31,32]. We analysed the presence of the endosomal marker caveolin-1 and clathrin on neutrophil-derived EVs by western blot and found both to be expressed (Fig. 5a). The endosomal marker caveolin-1 is known to be involved in vesicle exocytosis. Specifically blocking exocytosis by neutrophil pretreatment with BAPTA-AM significantly reduced the generation of neutrophil-derived EVs (Fig. 5b). The vesicular coating molecule clathrin plays a well characterized role in vesicle internalization into living cells. Blocking clathrin-dependent vesicle internalization with chlorpromazine significantly decreased the production of $TxA_2$, indicating decreased EV uptake into platelets (Fig. 5c). Western blot analysis of neutrophil-derived EVs also revealed the presence of the adhesion molecule Mac-1 on these vehicles (Fig. 5d). Pretreatment of neutrophil-derived EVs with a blocking Mac-1 antibody significantly decreased the production of $TxA_2$, indicating decreased EV uptake into platelets (Fig. 5e). The combination of chlorpromazine treatment and a blocking Mac-1 antibody did not show an additional effect in platelet binding or $TxA_2$ production, indicating that both Mac-1 and clathrin cooperate for the uptake of neutrophil-derived EVs by platelets (Fig. 5e,f). To further prove that EVs are internalized by platelets, isolated neutrophil-derived EVs were co-incubated with platelets, followed by isolation of washed platelets. Western blot analysis of these platelet preparations demonstrated the presence of Mac-1 in platelet lysates after co-incubation with neutrophil-derived EVs (Fig. 5g). Finally, we investigated whether the inhibition of Mac-1 dependent neutrophil-derived EV uptake into platelets would also affect survival during *E. coli*-induced pneumonia *in vivo*. GPIbα antibody-treated animals, which are susceptible to bacterial spread (Fig. 4i), were reconstituted with neutrophil-derived EVs that had been incubated with or without Mac-1 antibody. While reconstitution with untreated EVs improved survival, pretreatment of EVs with a blocking Mac-1 antibody before reconstitution did not affect survival after induction of pneumonia (Fig. 5h). Thus, neutrophil-derived EVs are essential for immune protection of infected lungs by acting on platelets. Previous reports on reverse transcellular communication by EV transport from activated platelets to neutrophils imply a possible Cox1 shuttling from platelets to neutrophils. However, we investigated the activity of Cox1 after co-incubation of isolated WT neutrophils with either WT or $Cox1^{-/-}$ platelets and could not detect a significant difference in neutrophils (Fig. 5i). Interestingly, the reconstitution of GPIbα-treated animals with isolated EVs improved survival after induction of pneumonia (Fig. 4n). We induced pneumonia in GPIbα-treated animals that were consecutively reconstituted with isolated EVs that had been treated with the GPIbα blocking antibody *in vitro* before re-injection into the animals. The direct GPIbα blockade prevented the rescue of the phenotype and led to impaired neutrophil recruitment and increased bacterial c.f.u. numbers in the BAL, lung and spleen of these animals compared with GPIbα-treated animals which received untreated, isolated EVs (Supplementary Fig. 6a–d). Thus, we assume that the amount of GPIbα antibody injected into the recipient animals to prevent endogenous EV generation did not suffice to also completely block the internalization of the isolated EVs which were re-injected into these animals.

**Cox1 modulates neutrophil recruitment during pneumonia.** Thromboxane $A_2$ production during pneumonia leads to increased endothelial ICAM-1 expression[9]. Expression of ICAM-1 at the mRNA and protein level in isolated murine lung microvascular endothelial cells (MLMVEC) was increased after exposure to stimulated neutrophils and platelets, and this required the expression of Cox1 in platelets, but not neutrophils (Fig. 6a,b). We confirmed these findings by ICAM-1 immunofluorescence analysis in MLMVECs (Fig. 6c). In agreement with this observation, immunofluorescence staining of lung sections from WT and $Cox1^{-/-}$ mice showed decreased ICAM-1 expression in $Cox1^{-/-}$ mice during pneumonia (Fig. 6d). To investigate the ICAM-1-dependent migration of neutrophils through the lung, we performed confocal microscopy of viable lung sections *ex vivo* (Fig. 6e; Supplementary Movies 1 and 2). Neutrophils in lungs from $Cox1^{-/-}$ mice and in lungs from WT animals pretreated with a blocking ICAM-1 antibody showed a significantly reduced migration velocity (Fig. 6f) and distance relative to untreated

**Figure 4 | EV-mediated shuttling of arachidonic acid into platelets is necessary for host defence.** Isolated PMNs, PLTs or both were pretreated with blocking antibodies against P-selectin (clone RB40.34, $5 \mu g \, ml^{-1}$) or GPIbα (clone Xia.B2, $5 \mu g \, ml^{-1}$) and stimulated with ADP (10 μM) and fMLP (10 μM) at 37 °C for 30 min. (**a**) EVs were quantified in the supernatant ($n = 4$). (**b**) The arachidonic concentration in the EV fraction was quantified by ELISA ($n = 4$). (**c**) Uptake of fluorescently-labelled, isolated neutrophil EVs (labelled with a green fluorescent cell tracker) in isolated platelets labelled with a CD41-PE antibody (dilution 1:200) was analysed by confocal microscopy (exemplary micrograph, scale bar equals 10 μm). (**d**) Uptake of neutrophil-derived EVs into platelets (exemplary micrograph, scale bar equals 10 μm). (**e**) Colocalization of EVs and Cox1 in platelets (exemplary micrograph, scale bar equals 5 μm). PMNs and platelets were isolated from WT mice and treated with a blocking antibody against GPIbα. (**f**) $TxB_2$-$C^{14}$ in control and ADP (10 μM)/fMLP (10 μM)-stimulated samples after pretreatment with a blocking GPIbα-antibody (clone Xia.B2, $5 \mu g \, ml^{-1}$) with or without substitution of isolated EVs ($n = 3$). (**g**) Total $TxB_2$ production in control and ADP/fMLP-stimulated samples ($n = 3$). (**h**) Cox1 activity after treatment with isotype, blocking antibody against GPIbα (clone Xia.B2, $5 \mu g \, ml^{-1}$) or blocking GPIbα antibody and isolated EVs ($n = 3$). Wild-type mice pretreated with the isotype or blocking antibody against GPIbα (clone Xia.B2, 50 μg per mouse) were injected intratracheally with viable *E. coli* and (**i**) the amount of circulating platelet-neutrophil aggregates in the blood, (**j**) neutrophil recruitment into the alveoli and the c.f.u. count in the BAL (**k**), lung tissue (**l**) and the spleen (**m**) were analysed after 24 h ($n = 4$). (**n**) Wild-type mice were pretreated with GPIbα blocking antibody (clone Xia.B2, 50 μg per mouse) and received isolated neutrophil EVs or control and survival was assessed 24 h after instillation of $8 \times 10^6$ viable *E. coli* ($n = 7$–8). Isolated PMNs and platelets were pre-incubated with a blocking antibody against GPIbα (clone Xia.B2, $5 \mu g \, ml^{-1}$) or isotype control and stimulated with ADP (10 μM) and fMLP (10 μM) at 37 °C for 30 min. The concentration of LTB4 (**o**) and LTC4 (**p**) was measured in the supernatant ($n = 4$). Mean ± s.e.m., ANOVA plus Bonferroni testing, two-tailed *t*-test in 4b, log rank test in 4n *$P < 0.05$.

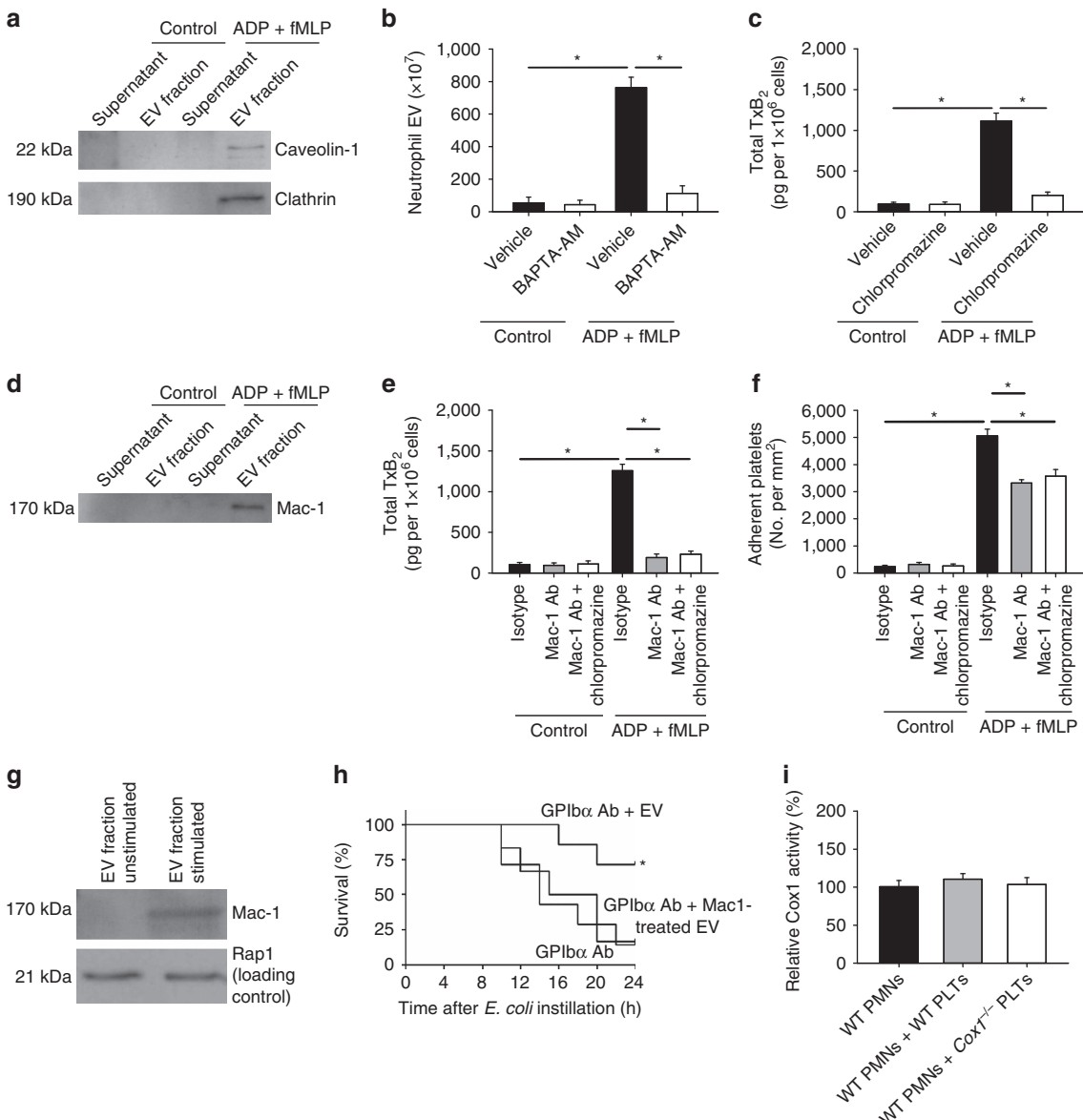

**Figure 5 | EV shuttling involves directed release and uptake mechanisms.** (**a**) Caveolin-1 and Clathrin in supernatant and EV fraction from control and ADP (10 μM)/fMLP (10 μM)-stimulated samples was detected by western blot (exemplary blot from three experiments). (**b**) Isolated platelets and neutrophils (ration 1:10) were co-incubated and control samples and stimulated samples were pretreated with vehicle or 5 μM BAPTA-AM and the number of generated EVs was quantified ($n = 3$). (**c**) Total TxB$_2$ production in control and ADP (10 μM)/fMLP (10 μM)-stimulated samples after pretreatment with vehicle or 10 μg ml$^{-1}$ chlorpromazine ($n = 3$). (**d**) Mac-1 (CD11b) in supernatant and EV fraction from control and ADP (10 μM)/fMLP (10 μM)-stimulated samples was detected by western blot (exemplary blot from 3 experiments). (**e**) Total TxB$_2$ production in control and ADP (10 μM)/fMLP (10 μM)-stimulated samples after pretreatment with a blocking Mac-1 antibody (clone M1/70, 5 μg ml$^{-1}$) or antibody plus 10 μg ml$^{-1}$ chlorpromazine ($n = 3$). (**f**) Platelet adhesion in fibrinogen-coated flow chambers in control and ADP/fMLP-stimulated samples after pretreatment with a blocking Mac-1 antibody (clone M1/70, 5 μg ml$^{-1}$) or antibody plus 10 μg ml$^{-1}$ chlorpromazine ($n = 3$). (**g**) Western Blot of Mac-1 in platelets co-incubated with EV fraction from unstimulated and stimulated neutrophils (exemplary blot from three experiments). (**h**) Wild-type mice were pretreated with GPIbα blocking antibody (clone Xia.B2, 50 μg per mouse) and received isolated neutrophil EVs pretreated with or without a blocking Mac-1 antibody. Survival was assessed 24 h after instillation of $8 \times 10^6$ viable E. coli ($n = 6-7$). (**i**) Cox1 activity in WT neutrophils alone or after co-incubation with activated WT or $Cox1^{-/-}$ platelets ($n = 5$). Mean ± s.e.m., ANOVA plus Bonferroni testing, log rank test in 5 h *$P < 0.05$.

controls (Fig. 6g). To investigate whether ICAM-1 contributes to neutrophil recruitment and bacterial clearance in our model of E. coli-induced pneumonia, mice were injected with a blocking anti-ICAM-1 antibody before inducing pneumonia. Blocking ICAM-1 caused a significantly decreased number of neutrophils in the BAL after instillation of E. coli (Fig. 6h) and a significantly increased c.f.u. count in the BAL, lung and spleen with no additional effect when ICAM-1 was blocked in $Cox1^{-/-}$ mice (Fig. 6i–k).

## Discussion

We have demonstrated that the interaction of platelets with neutrophils is required for enhancing thromboxane production from activated platelets. Cox1 in platelets was required for the production of thromboxane A$_2$ using predominantly neutrophil-derived arachidonic acid as a substrate, and the physical proximity between platelets and neutrophils led to enhanced Cox1 activity. Mechanistically, neutrophil-derived

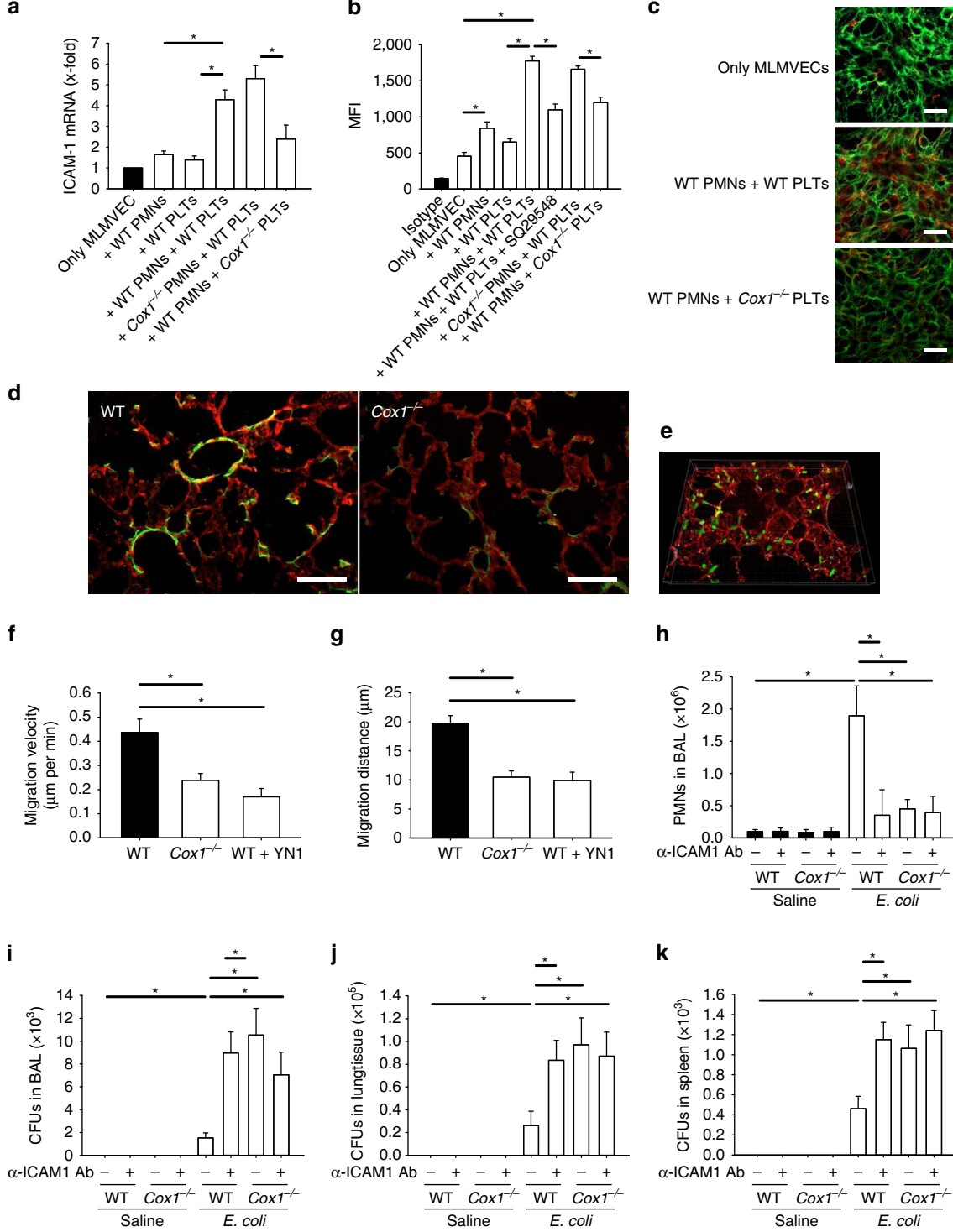

**Figure 6 | Cox1 modulates neutrophil recruitment during pneumonia.** MLMVEC were isolated from WT mice and co-incubated with platelets and PMNs and ICAM-1 mRNA expression (**a**) and ICAM-1 surface expression (**b**) were analysed ($n = 4$). (**c**) Exemplary immunofluorescence staining of ICAM-1 on the surface of MLMVEC after co-incubation with platelets and PMNs (scale bar equals 50 μm). (**d**) ICAM-1 (green) and PECAM-1 (red) immunofluorescence staining in fixed lung sections from WT and $Cox1^{-/-}$ mice after instillation of *E. coli* (scale bar equals 100 μm). WT mice or $Cox1^{-/-}$ mice received a blocking ICAM-1 antibody (clone YN1, 50 μg per mouse) or isotype control and were injected intratracheally with viable *E. coli* ($6 \times 10^6$ c.f.u.'s per mouse). (**e**) Exemplary 3D confocal image of *E. coli* infected lung samples stained against PMNs (clone RB6-8C5, green) and PECAM (clone 390, red). Migration velocity (**f**) and distance (**g**) of PMNs in WT mice, $Cox1^{-/-}$ mice and WT mice after pretreatment with a blocking ICAM-1 antibody (clone YN1, 50 μg per mouse) ($n = 3$). (**h**) Neutrophil recruitment into the alveoli and the c.f.u. count in the BAL (**i**), lung tissue (**j**) and the spleen (**k**) were analysed after 24 h ($n = 4$). Mean ± s.e.m., ANOVA plus Bonferroni testing, *$P < 0.05$.

arachidonic acid was shuttled from neutrophils to platelets by EVs, and this process required the binding of GPIbα on platelets after the initial P-selectin-mediated contact. EV liberation from neutrophils and uptake by platelets was found to be a directed process involving clathrin and Mac-1. Furthermore, we showed that Cox1-mediated thromboxane production from platelets is required for sufficient neutrophil recruitment into the lung and efficient bacterial clearance during bacterial pneumonia. In addition, thromboxane production was required for increased ICAM-1 expression on endothelial cells.

Platelets are a well-recognized part of the innate immune response to pathogenic infections[6,33]. Platelets possess a rich intravascular storage of pro-coagulatory and pro-inflammatory mediators, but are also capable of synthesizing lipid mediators[33]. Thus, platelets are involved in numerous inflammatory processes, such as acute lung injury, renal ischemia–reperfusion injury and sepsis[8,9,13,19,34,35]. In addition, platelets have been shown to be an element of immune surveillance against invading pathogens in the liver[36]. Here, we provide experimental evidence that platelets rely on neutrophils to supply arachidonic acid substrate for maximal thromboxane generation during inflammatory conditions, which is in line with previous reports[37]. Maugeri and colleagues discovered that prostaglandin synthesis is dependent on neutrophils providing arachidonic acid to platelets[38] and that the close proximity of platelets and neutrophils allows for the transcellular metabolism of eicosanoids between the cells[37]. Furthermore, it is known that the bilateral exchange of metabolites is required for the production of LTB4 and LTC4 (ref. 39). However, molecular mechanisms responsible for transcellular lipid metabolism have remained unexplained. Using an approach with radioactive-labelled arachidonic acid, we demonstrated that efficient platelet thromboxane generation requires eicosanoid metabolites originating from neutrophil arachidonic acid.

We show now that a mechanism for the transcellular communication between neutrophils and platelets involves EVs and that the EV-mediated metabolite shuttling requires GPIbα and P-selectin. Furthermore, we demonstrate that the EV release by neutrophils and their uptake into platelets is a directed process. We find the endosomal marker caveolin-1 to be expressed on neutrophil-derived EVs. This finding is supported by previous studies reporting the presence of caveolin-1 on EVs from melanoma cells in tumour patients[40]. Caveolin-1 also has an important role in immune cell function and host defence[32]. In particular, neutrophil caveolin-1 is involved in the pathogenesis of pneumonia[41,42]. However, the role of caveolin-1 in EV substrate shuttling between neutrophils and platelets has yet to be investigated. Our data also suggest the presence of the vesicular coating molecule clathrin as well as Mac-1 on neutrophil-derived EVs and the functional importance of both molecules for the uptake and internalization into platelets. This data is in line with a previous study reporting that macrophage-derived EVs are internalized into placental cells by clathrin-dependent endocytosis inducing pro-inflammatory cytokine release[43]. This is, to our knowledge, the first report indicating that intercellular transport of metabolic substrates is specifically mediated by EV shuttling between neutrophils and platelets, and not the other way from platelets to neutrophils. While we could identify this route of intracellular substrate transport form neutrophils to platelets in vitro, we cannot fully rule out that the vice versa route from platelets to neutrophils may also play a role in vivo. Indeed, it has been reported that platelet microparticles may be internalized into neutrophils in a 12(S)-HETE-dependent manner during inflammatory arthritis[44]. Previous studies also indicate that opsonized bacterial particles may induce the release of neutrophil-derived EVs with antibacterial properties[45]. Furthermore, different stimuli may induce the generation and release of different EV subsets with divergent molecular composition and antibacterial functions[46]. Interestingly, we did not detect significant alterations in the concentration of EVs after E. coli pneumonia induction between control and platelet-depleted mice. This may be explained by the fact that EV shuttling from neutrophils to platelets represents a process in the immune cell recruitment taking place at the emigration of neutrophils from the microvasculature into the lung tissue, eventually enabling the neutrophils to enter the alveolar space. Another explanation may be that different mechanisms of EV generation in the alveolar compartment and secretion into the BAL are involved apart from platelet-neutrophil interactions. The exact temporal-spatial contribution of EV shuttling to the distinct process of neutrophil recruitment has yet to be investigated in detail to gauge the role of detectable neutrophil-derived EVs in the bronchoalveolar lavage fluid.

Research over the past several years has identified an important role for cyclooxygenases and their metabolites in inflammatory processes[47–52]. However, to this date it was unknown which conversion enzyme in platelets is necessary for utilization of neutrophil eicosanoid metabolites to booster platelet thromboxane generation. In our study, we identified Cox1 as the prevalent cyclooxygenase responsible for thromboxane generation using neutrophil-derived arachidonic acid as a substrate. This is in line with a recent report demonstrating that the platelet Cox1 signalling pathway regulates leukocyte activation and affects disease progression during systemic inflammatory disorders[53]. On first glance it may appear surprising that platelets depend on arachidonic acid from neutrophil-derived EVs to efficiently generate thromboxane, as they possess cellular arachidonic acid storages themselves. However, this phenomenon may be explained by our finding that the uptake of neutrophil-derived EVs not only provides substrates to platelets, but also increases Cox1 activity in platelets, thus optimizing and synchronizing the usage of substrates for thromboxane generation.

Interestingly, the physical contact between platelets and neutrophils also influences cell-autonomous functions of neutrophils. It is known that platelet engagement causes the upregulation of neutrophil adhesion molecules and increases ROS production[54–56]. While the exact mechanism of these interactions in vivo are still unclear, it is possible that platelets use the neutrophil's uropod, a PSGL-1-enriched microdomain that protrudes into the luminal space, as a means for efficient binding to neutrophils that first accumulate in the inflamed lungs[14]. The current study establishes that one relevant mechanism of intercellular communication within this structure is through lipid exchange enabled by EVs. Transcellular metabolism from platelets to neutrophils is also of importance, as neutrophils require hydroxyeicosatetenoic acid (12-HETE) produced by platelet-specific lipoxygenases to generate leukotrienes[57]. Interestingly, this metabolic pathway also requires the binding of platelet P-selectin to neutrophil PSGL-1, and this physical interaction regulates the activity of the conversion enzyme LTC4 synthase further downstream in neutrophils[37,58]. Thus, transcellular metabolism between platelets and neutrophils seems to be a bidirectional process with benefits for both cell types[59]. It has been shown that the balance in this system of transcellular eicosanoid metabolism is also affected by the activity of the respective conversion enzymes in platelets and neutrophils. For example, a decrease in neutrophil lipoxygenase activity results in a significant increase in platelet thromboxane $B_2$ by increased supply and shift of eicosanoid substrates from neutrophils to platelets in their close proximity[60].

Prostaglandins are among the most prominent lipid mediators released from platelets with the pro-inflammatory thromboxane being an important mediator from this group. In this study, we

present evidence that thromboxane generated by platelets from neutrophil substrates is of crucial importance for leukocyte recruitment and bacterial clearance during septic infections. This observation is in accordance with numerous reports demonstrating the importance of thromboxane for the development of an immune response against invading pathogens. For example, thromboxane has been implicated in the governance of endothelial cell inflammation[61], in the regulation of vascular permeability changes during inflammatory conditions in the lung[62] as well as in the recruitment of immune cells to the focus of infection[9,63].

In summary, we have identified a role for transcellular prostaglandin metabolism between neutrophils and platelets, mediated by metabolite shuttling via EVs and involving Cox1, which affects neutrophil recruitment and bacterial killing. These findings are of clinical importance, as the inhibition of cyclooxygenases is a common therapeutic approach for pain relief and anti-inflammatory therapy. In addition, the cyclooxygenase inhibitor acetylsalicylic acid (aspirin) is widely used for platelet inhibition in patients after cardiac bypass grafting. Thus, inhibitory effects on neutrophil recruitment may be of great importance in these patient collectives. Furthermore, the potential use of cyclooxygenase inhibitors could be of potential interest for the therapy of inflammatory disorders involving uncontrolled neutrophil recruitment and warrants further research in this field.

## Methods

**Animals and reagents.** We used 8–12-week-old male C57BL/6 mice, $Cox1^{-/-}$ mice (Taconic, Hudson, NY, USA), and $Cox2^{-/-}$ mice (Taconic). The mice were kept in a barrier facility under specific pathogen-free (SPF) conditions. All animal experiments were approved by local government authorities and were in agreement with the National Institutes of Health Guide for the Care and Use of Laboratory Animals. Unlike otherwise stated, all reagents were obtained from Sigma-Aldrich (Taufkirchen, Germany).

**Murine pneumonia model.** We used a murine model of E. coli induced pneumonia as published by Ittner and colleagues[64]. Overnight cultures (37 °C) of E. coli (ATCC strain 25922) were grown in Tryptic Soy medium, washed and resuspended in sterile saline (0.9%). Mice were anaesthetized by intraperitoneal injection of ketamine (125 µg g$^{-1}$ body weight; Pfizer, New York, USA) and xylazine (12.5 µg g$^{-1}$ body weight; Bayer, Leverkusen, Germany). We used two different sets of experiments. For the first set of experiments, animals were challenged with $6 \times 10^6$ viable E. coli per mouse. At this inoculation dose, all mice survived the 24 h observation period. In a separate set of experiments, mice were challenged with a higher inoculation dose ($8 \times 10^6$ viable E. coli per mouse) which allowed the survival analysis. After 24 h, the mice challenge with $6 \times 10^6$ viable E. coli were sacrificed and the lungs were lavaged five times with 0.7 ml physiologic saline solution. The number of neutrophils in the BAL was counted using kimura staining. Neutrophils were counted using an improved Neubauer counting chamber and an inverted cell culture microscope (Primovert, Carl Zeiss, Göttingen, Germany) equipped with a $10 \times 0.75$ NA objective. A total of 4 fields with 16 standardized subfields in each individual field were counted for each sample. For some experiments platelet depletion in WT mice was achieved by intraperitoneal administration of busulfan as described previously[8,65]. The application of busulfan decreases platelet counts by >90% (ref. 8). PMNs were depleted in some mice by injection of anti-Ly6G antibody (clone 1A8, 200 µg per mouse i.p., Biolegend). Colony-forming units in the lung, blood and spleen were counted by serial plating on TSA agar plates[64]. For the Klebsiella pneumoniae-induced pneumonia model, mice were intratracheally instilled with $2 \times 10^7$ viable K. pneumoniae (ATTC strain 13883).

**Intravital microscopy of the lung.** Intravital microscopy of the lung was performed as described previously[8,66]. Briefly, animals were anaesthetized and mechanically ventilated. A thoracotomy was performed to expose the right middle lung lobe and the lung was hold in position using a custom-built fixation device with an integrated observation window[66]. Immediately before imaging animals were injected with an Alexa488-coupled anti-Gr1 antibody (clone RB6-8C5, 5 µg per mouse, purified from hybridoma supernatant) and a PE-coupled anti-CD41 antibody (clone MWReg30, 5 µl per mouse, BD Biosciences, Franklin Lakes, NJ, USA). High-speed multichannel fluorescence microscopy was performed on an upright microscope (Axioskop; Carl Zeiss, Göttingen, Germany) equipped with a Lambda DG-4 ultra high speed wavelength switcher (Sutter Instruments, Novato, CA, USA) and a $40 \times 0.75$ NA saline immersion objective. Videos were recorded

with a digital camera (Sensicam QE) and analysed with Slidebook Software (Version 5; Intelligent Imaging Innovations, Göttingen, Germany).

**Quantification of platelet–neutrophil interactions in vivo.** Whole blood samples were withdrawn from mice and stained with Alexa633-coupled anti-Gr1 antibody (clone RB6-8C5, dilution 1:100), PE-coupled anti-CD41 antibody (clone MWReg30, dilution 1:100, BD Biosciences), FITC-coupled anti-Ly6B2 antibody (clone 7/4, dilution 1:100, AbD Serotec, Düsseldorf, Germany), and PerCP-coupled anti-CD45 antibody (clone 30-F11, dilution 1:100, BD Biosciences). Platelet-neutrophil aggregates were quantified by measuring the percentage of CD41$^+$ neutrophils (CD45$^+$Gr1$^+$7/4$^+$) using a flow cytometer (BD FACSCanto II, BD Biosciences).

**Thromboxane determination by radioimmunoassay.** Thromboxane was quantitated using a radioimmunoassay. Neutrophils and/or platelets (ratio 1:10) were loaded with C$^{14}$-labelled arachidonic acid (Perkin Elmer, Waltham, MA, USA). After stimulation with 10 µM ADP and 10 µM fMLP at 37 °C for 30 min, C$^{14}$-TxB$_2$ was immunoprecipitated from the supernatant and detected by measuring the ß-radiation activity using a TRI-CARB 2900 TR liquid scintillation counter (Packard Instruments, Meriden, CT, USA). Total TxB$_2$ levels were measured using an ELISA kit according to the manufacturer's instructions (R&D Systems, Minneapolis, MN, USA).

**EV preparation and quantification.** For the generation of isolated neutrophil EVs the cells were stimulated with fMLP and EVs were purified from the supernatant using the ExoQuick-TC kit and EV quantification was performed using the Exocet kit according to the manufacturer's instructions (System Biosciences, Mountain View, CA, USA). Arachidonic acid in purified neutrophil EVs was measured by ELISA (R&D Systems). For reconstitution experiments, isolated neutrophils ($\sim 40 \times 10^6$) from wild-type bone marrow were stimulated with fMLP. EVs ($\sim 1 \times 10^8$) were isolated from the supernatant and reinjected into mice 15 min after intratracheal instillation of E. coli. For in vitro assays with EV reconstitution, neutrophil-derived EVs were added at a concentration of $2 \times 10^7$ EVs per ml. To further characterize the investigated extracellular vesicles, we analysed our study material according to the guidelines published by the International Society of Extracellular Vesicles[67]. As the concomitant contamination of extracellular vesicle preparations with apoptotic cell bodies may obscure the studied effects of extracellular vesicles, we analysed stained EV supernatants with Trypan Blue staining and could not detect significant amounts of contamination with apoptotic cell remnants in our EV preparation. As suggested by the International Society of Extracellular Vesicles we also investigated the expression of proteins that neutrophil-derived EVs may inherit from their origin cell (that is, neutrophils). Western blot analysis revealed that the integrins Mac1, the integrin subunits CD11a ($\alpha_L$) and CD18 ($\beta_2$) as well as PSGL-1 (CD162) are present in lysates from neutrophil-derived EV preparations, all of which are also expressed on the membrane of neutrophils, and the commonly EV-associated molecule CD63 (Supplementary Fig. 7a). In contrast, the histones H2A and H3 could not be detected in lysates from neutrophil-derived EV preparations, as these proteins are usually nucleus-associated and do not associate with endosomal structures. To also show the functional dose-response relationship of the neutrophil-derived EV isolates we blocked endogenous EV liberation in a stimulated platelet-neutrophil co-culture in vitro and substituted isolated EVs at different concentrations to measure the amount of TxB$_2$ in this system and could observed increasing TxB$_2$ generation with increasing substitution amounts of isolated neutrophil-derived EVs (Supplementary Fig. 7b).

**Western blot.** Isolated EVs were lysed with RIPA buffer. Lysates were boiled with Lämmli sample buffer, run on 10% SDS–PAGE gels and immunoblotted using antibodies against Caveolin-1 (Cell Signaling Technology, Danvers, MA, USA, Cat. No. 3238, dilution 1:1,000), Clathrin (clone P1663, dilution 1:1,000, Cell Signaling Technology) or Mac-1 (clone M1/70, dilution 1:1,000, Biolegend, San Diego, CA, USA). Immunoblots were developed using an ECL system (GE Healthcare, Little Chalfont, UK). Uncut western blots are supplied in Supplementary Fig. 8.

**Platelet flow chamber.** Rectangular glass capillaries ($20 \times 200$ µm) were coated with fibrinogen (1,500 µg ml$^{-1}$) for 2 h followed by blocking of unspecific binding sites with casein 1% (Thermo Fisher Scientific, Waltham, MA, USA) for 1 h. One end of the glass capillary was connected to a PE50 tubing (BD Biosciences) and used to control the wall shear stress in the capillary. Wall shear stress was adjusted to $\sim 20$ dyne cm$^{-2}$ resembling arterial flow conditions. Anticoagulated whole-blood samples were withdrawn from WT mice and samples were stimulated with 10 µm ADP and 10 µm fMLP at 37 °C for 5 min. The chamber was perfused for 2 min and washed with PBS for 1 min. Representative fields of view were recorded using an SW40/0.75 objective and a digital camera. The numbers of adherent platelets per mm$^2$ were calculated.

**Transmission electron microscopy.** For electron microscopy the lung was perfused with PBS via the right ventricle, followed by 2% glutaraldehyde, 2%

paraformaldeyde in 0.2 M cacodylate buffer, pH 7.4. The lung was removed and small samples were further fixed under low vacuum until they settled down. The specimen was post-fixed with 1% osmiumtetroxide and 1.5% potassium ferrocyanide, dehydrated and embedded in epon. Sixty nanometer ultrathin sections were cut (Leica UC6 ultramicrotome, Vienna, Austria) and counterstained with uranyl acetate and lead. Samples were inspected on a transmission electron microscope at 80 kV (Fei-Tecnai12, FEI, Eindhoven, Netherlands) and pictured with a CCD camera (Megaview, SIS, Münster, Germany).

**Cox activity assay.** Cox1 activity in supernatants was analysed according to the manufacturer's protocol (Cayman Chemical, Ann Arbor, MI, USA). Cox2 activity was pharmacologically blocked by use of the specific inhibitor DuP-697.

**PMN migration ex vivo.** The Migration of PMNs in lung tissue ex vivo was investigated using a technique published by Hasenberg and colleagues, with some modifications[68]. Mice were injected i.t. with E. coli. After 4 h, animals were injected with Alexa488-coupled anti-Gr1 antibody (clone RB5-8C6, 5 µg per mouse) and Alexa568-coupled anti-PECAM (clone 390, 50 µg per mouse, BD Biosciences) to stain neutrophils and endothelial cells, respectively. Mice were sacrificed and lungs were filled with 1 ml of low-melting agarose. After removal, lungs were cut using a vibratome. Lungs were fixed in a cell culture dish, submersed in PBS and time-lapse z-stacks were recorded using a spinning disc confocal microscope (CellObserver SD, Zeiss, Göttingen, Germany) equipped with a 20 × /1.0 NA objective. Neutrophil migration velocity and displacement were analysed using FIJI[69].

**MLMVEC isolation and analysis of ICAM-1 expression.** Murine lung micro-vascular endothelial cells (MLMVEC) were isolated and cultured from murine lungs. Lungs were mechanically minced and enzymatically digested with 20 mg ml$^{-1}$ collagenase A (Roche, Basel, Switzerland) at 37 °C for 2 h. Dynabeads (Life Technologies, Carlsbad, CA, USA) were incubated with anti-PECAM-1 antibody (clone Mec13.3, 30 µg per 200 µl, purified from hybridoma supernatant) at 4 °C for 24 h. Lung homogenisates were incubated with dynabeads at 4 °C for 45 min, followed by magnetic separation of the dynabeads from the supernatant. MLMVEC cells were culture on gelatin-coated dishes for 4–7 days. The mRNA from MLMVEC was extracted using Trizol extraction (peqGOLD TriFast, Peqlab, Erlangen, Germany). mRNA was transcribed into cDNA using M-MLV reverse transcriptase. Quantitative expression analysis by qRT-PCR was performed using primers specific for murine ICAM-1 and GAPDH[70]. Surface expression of ICAM-1 was analysed using immunofluorescence staining and flow cytometry as described previously[9].

**Immunofluorescence staining.** Platelet/neutrophil co-cultures isolated neu-trophils and platelets were immobilized on poly-L lysine coated plastic dishes (8-well slide, Ibidi) and fixed with 4% paraformaldehyde. After blocking, samples were incubated with antibodies against Ly6G (clone 1A8, dilution 1:100, Biole-gend), CD49b (clone H2Mα, dilution 1:100, Biolegend), LBPA (clone 6C4, dilution 1:100, Echelon Biosciences) and Cox1 (clone H-62, dilution 1:100, Santa Cruz Biotechnologies) at 4 °C overnight. Secondary staining was performed with Alexa-Fluor coupled secondary antibodies (all from Invitrogen). Hoechst 33442 was used for nuclear counterstaining. Images were acquired using a LSM780 confocal microscope (Carl Zeiss) equipped with a 63 × immersion objective. Images were analysed using ZEN 2 software (Carl Zeiss).

**Statistics.** Statistical analysis was performed with SPSS (version 22.0) using Wilcoxon-test or t-test as appropriate. More than two groups were compared using one-way ANOVA followed by Bonferroni testing. Data distribution was assessed using Kolmogorov–Smirnov test or Shapiro–Wilks test. Survival analysis was performed using log-rank test. All date are represented in means ± s.e.m. A P value <0.05 was considered as statistically significant. For in vivo experiments, the provided n is the number of animals used per experiment. For in vitro experiments, n describes the number of independent experiments, each done at least in technical triplicates.

**Data availability.** The data that support the findings of this study are available from the corresponding author upon reasonable request.

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

## Acknowledgements

The authors would like to thank Nadja Giesbrecht and Mareike Schlüter for expert technical support. This work was supported by the Deutsche Forschungsgemeinschaft (ZA428/6-1 and ZA428/8-1 to A.Z., RO 4537/2-1 to J.R.) and Cells-in-Motion Cluster of Excellence EXC 1003-CiM (University of Münster, Germany; to A.Z.). Grant SAF2012-31142 from MINECO (to A.H.). Grant HL107386 from the NHLBI (to M.R.L.). The Centro Nacional de Investigaciones Cardiovasculares (CNIC) is supported by the MINECO and the Pro-CNIC Foundation.

## Author contributions

J.R. performed the experiments, analysed the data and wrote the manuscript; K.K. and J.S. performed experiments; H.V.A., M.R.L. and A.H. interpreted data and contributed to writing the manuscript; A.Z. conceived of the study, analysed the data and wrote the manuscript.

## Additional information

**Competing financial interests:** The authors declare no competing financial interests.

