## [Peer Review File · Nature Communications]

Reviewers' comments:

Reviewer #1 (Remarks to the Author):

This is a very comprehensive study that elucidates mechanism of transcellular synthesis via exosomes between platelets and neutrophils. I have three major issues that need to be resolved with additional experiments.

1) How is P-selectin upregulated on platelets in vivo in infections? Please demonstrate that this actually occurs. Figure 1 shows that the effects of fMLP and ADP are blocked by P-selectin antibody in a test tube but these effects may not be occurring in vivo. Why did the authors not add E.coli rather than ADP and fMLP? One clear issue is that while ADP causes P-selectin expression E.coli and/or LPS do not activate P-selectin very well. This reviewer is not doubting the data, simply the disconnect between in vivo and in vitro experiments.

2) Blocking P-selectin in vivo reduced the number of neutrophils recruited into BAL but this does not necessarily invoke a role for platelets. Bone marrow chimeras could delineate between platelets and endothelium.

3) It is strange that busulfan depletion of platelets prevented neutrophils from entering BAL. The authors need to provide some evidence that the busulfan does not affect neutrophil biology. In fact other approaches to depleting platelets would be important.

Reviewer #2 (Remarks to the Author):

The manuscript explores an interesting topic and the proposed exosome signaling mechanism is novel enough. The requirement of neutrophils and platelets in vivo shown in Figures 2 and 3 are convincing. However there are several major technical issues and over interpretations in experiments assessing the exosome mechanism. The authors make a strong case that platelets are important for neutrophil recruitment and that the TxB2 pathway is important in vivo. They also make a good case for neutrophils regulating the process. But the involvement of exosomes needs to be better analyzed and substantiated. Please see specific suggestions for Figures 1 and 4.

While most of the manuscript examines neutrophil recruitment in a convincing fashion, the NET data are weak and not convincing. NETs are not even mentioned in the abstract. It might be better to remove these data from the manuscript as it offers little and weakens the overall quality.

Finally the description of the experiments in the main text and legends is poor and in many cases (example figure 4) it is unclear how the experiments were performed. Also, the figure letters are lower case in the figures and capital in the text.

Specific comments:

Figure 1b is misinterpreted. When platelets are loaded with arachidonic acid they produce significant amounts of labeled TxB2. There is a two fold increase in radio labeled TxB2 which could be explained by having twice as many cellular sources. The data suggest that neutrophils might be able to supply arachidonic acid, but not "that neutrophils provide the majority of arachidonic acid required for maximal TxB2 production" (pg. 6). At this stage all we know from fig 1a is that both cells are needed and and from fig. 1b that both contribute. Perhaps comparing to the cells alone might help show that the neutrophils can produce TxB2 themselves which will then hint that they have to donate

arachidonic acid to platelets. Perhaps reordering the experiments might get around this problem.

Figure 1c and d. The labels are so small that they are impossible to read. Also, it is unclear which cells were labeled in these experiments. This is crucial because the data show that Cox enzymes are not important in neutrophils but only in platelets. Therefore, one can't use the mutants to decipher where the arachidonic acid is coming from as it would be the case if it were processed in neutrophils. The amount of radioactive signal based on Fig. 1b suggests that both cells are labeled here. Given the results here, it should only be neutrophils labeled in order to assess their transfer of arachidonic acid.

Figure 2i. It is not clear where the platelets are. Immunofluorescence microscopy is required.

Figure 4c. The authors need to substantiate the microscopy with biochemistry to show that exosomes are taken up by platelets. They can incubate platelets with exosome preparations from unstimulated and stimulated neutrophils, isolate away the platelets and show the incorporation of Mac-1 or other neutrophil exosome markers into platelets by WB.

Figure 4e-h. The authors use neutrophils here. However they should repeat these functional experiments with isolated exosomes to prove that they are sufficient to mediate the effect of neutrophils.

Supplementary figure 4c. These data are problematic. First, the legend states that red is staining for histone H3. This antibody should stain all nuclei then, not only NETs. But it does not. Why? The authors should use an antibody against citrullinated histone H3 instead. Furthermore, MPO is confusingly shown in yellow (!) in a merged image where green and red are present which makes it impossible to judge which marker is which. Overall it is impossible to deduce anything about NETs from these images other than there are fewer neutrophils present in Cox 1 $-/-$ mice which is of course not novel. The MPO/DNA complex ELISA alone is not sufficient to make the case for NETosis because secreted MPO could associate with bacterial or host DNA as well. Moreover, the effects based on these assays are not impressive.

Supplementary Figure 4e. The WT sample shows on structure that is stains for H3 (not citrullinated H3) but no other nucleus stains for H3. This is strange. Furthermore, the so called "NET" seems to lack DNA staining. Finally, only one potential "NET" is shown in an image of several intact neutrophils. The main text and legend do not specify how these neutrophils were activated. Only that they were incubated with activated platelets. According to Paul Kubers LPS activation of platelets to make NETs works in very few cells and only under flow conditions. Therefore, robust quantitation is needed to examine the effects of Cox deficiency in a system where the positive control is very poor to begin with. Furthermore, if LPS was used, it is odd that neutrophils do not appear polarised and activated.

Reviewer #3 (Remarks to the Author):

The work by Rossaint and colleagues relates to two interesting aspects; 1) interplay between platelets and neutrophils in the combat of bacterial infections in lungs, and 2) transcellular metabolism of arachidonic acid (AA) involving platelets and neutrophils during bacterial infection. For their study, they used two models of lung infections. Although several studies have reported evidence of collaborations between neutrophils and platelets for the generation of eicosanoids during infectious processes, one interesting aspect suggested in this present study is the involvement of neutrophil-derived exosomes. According to the authors, neutrophils release exosomes, which AA cargo is used by platelet machinery to synthesize thromboxane. Based on their observations, interactions between

platelets and neutrophils implicate P-selectin and PSGL-1. Furthermore, thromboxane induces endothelial cell expression of ICAM-1, thus permitting intravascular crawling and neutrophil recruitment at the site of bacterial infection. Ablation of this pathway (using KO mice, inhibitors, and blocking abs) leads to aggravated infection and reduction of mouse survival.

It is a very interesting study, containing a series of convincing and well-designed experiments. Some concerns require clarification:

Interactions between neutrophils and platelets, via P-selectin and PSGL-1 is not novel; it has been described by several groups, and some of the key studies have been cited by Rossaint et al. What is important in this study, however, is the transcellular metabolism of AA mediating thromboxane A₂ production. Transcellular metabolism of AA between platelets and other cellular lineages, including neutrophils, has been known for a long time (see for instance the work from Serhan C., Collier B., FitzGerald G., Boyce JA., and the more recent study from Levy BD in PNAS 2014 Nov 18;111(46):16526-31). The authors claim that the mechanism, revealed in their study, is novel and significant because it implicates neutrophil-derived exosomes. To prove this, they labeled AA with C14 in neutrophils or in platelets, and they could determine that most of the thromboxane produced transcellularly implicated AA from neutrophils and cox-1 from platelets.

1) One major issue here is that there is no experiment that excludes the (most plausible) possibility that platelet-derived cox-1 is actually shuttled to neutrophils via extracellular vesicles (e.g. microparticles), where it metabolizes neutrophil-derived AA. Such pathway has been reported in several studies (for instance, see Barry OP, JCI 1997 May 1;99(9):2118-27, and the more recent Duchez AC, PNAS 2015 Jul 7;112(27):E3564-73), which demonstrate shuttling of cox-1 between platelets and neutrophils. Hence, platelets and/or platelet microparticles seem to be internalized in neutrophils in Figure 4D,E.

2) A second issue regards the notion of intercellular communication via exosomes. The authors claim that exosomes, very specifically, are implicated in the transcellular metabolism of AA; however there is no experiment that confirms the involvement of exosomes, and rule out the implication of microvesicles or other types of extracellular vesicles. In fact, the marked differences between extracellular vesicles produced by cytoplasmic membrane budding and those stored in multivesicular bodies or apoptotic bodies were totally overlooked. Hence, the authors mistakenly state that "exosomes are also termed microvesicles or microparticles", which is absolutely incorrect (see introduction). Furthermore, although the authors used commercial kits to "isolate" and "enumerate" exosomes, it is well known that other particles (microparticles/microvesicles, ectosomes, apoptotic bodies, protein aggregates, secretory vesicles, etc.) can interfere in these kits and are in fact isolated together with exosomes, especially if no other gradient/velocity centrifugations are performed. The experiments in Figure 5 do not permit to confirm that the vesicles implicated are exosomes. The authors should refer to most up to date notions on extracellular vesicles and should follow recommendations from the international society of extracellular vesicles and the ISTH in their definition of exosomes and the design of their experiments (see Lotvall, JEV 2014 Dec 22;3:26913). They have to correctly define what material they are studying, and provide rationale for their definition.

3) When "exosomes" are utilized in their in vitro and in vivo studies, there is no indication of the concentrations used. Without this information, it is impossible to the reviewer to assess of the relevance of the experiments. Furthermore, extracellular vesicles in BAL during infection could be determined using appropriate approaches.

4) In Figure 1B, the authors mention that most of the thromboxane is synthesized from neutrophil-derived AA. However, this is based on the assumption that both platelets and neutrophils were equally labeled with C14-AA, and that C14-AA is distributed equally in phospholipid classes in both cellular lineages. Since important quantities of TxB₂ also originate from platelet-derived AA, it is important to

confirm this authors' statement.

5) In addition, it is surprising that platelets, which are recognized for their important pool of membrane, require AA supply from other cells. Thromboxane synthase is inactivated by its products (a process known as suicide inactivation), and substrate availability in platelets is generally not limiting. The authors should comment on this.

6) The authors rapidly rule out IIbIIIa in neutrophil-platelet interactions, using an in vitro system where fibrinogen is absent. Such assay should be performed in presence of plasma, and further confirmation in vivo will be needed to exclude a contribution of IIbIIIa.

7) Although the assay that revealed transcellular metabolism of AA between platelets and neutrophils is consistent with in vivo data implicating thromboxane (using antagonist SQ29548), more information on the complete set of eicosanoids produced between neutrophils (exosomes) and platelets would be informative. For instance, what do we know on the levels of lipoxin A4, cystenyl leukotrienes in this system?

Minor

1) Supplemental Figure 2 comes before supplemental Figure 1

2) It should be indicated when BAL and blood were collected after bacteria instillation. In conditions where mouse survival was impaired, were the fluids collected prior or after death? Were all the mice used for the generation of the data (BAL, CFU, TxB2 etc.), or only those that (did not) survived?

3) Some graphs are hard to read. See Figures 1C and D for instance.

4) In the result section (Figure 5): "vehicle exocytosis" or vesicle exocytosis?

5) Details on methods are frequently lacking. For instance, neutrophils in BAL were quantified using kimura staining, but there is no indication on the number of fields and cells counted. More information on this approach would be needed.

Reviewers' comments:

Reviewer #1 (Remarks to the Author):

The authors did a great job of addressing my concerns this can be accepted.

Reviewer #2 (Remarks to the Author):

The authors have fully addressed all of my comments.
In my view the story is now very clear, experimentally sound well rounded.

Reviewer #3 (Remarks to the Author):

This is an improved version of their manuscript, with several additional experiments and clarifications. There are lingering concerns, however, that still need to be addressed.

1. One key concept that is brought by the authors is the directed EV shuttling "from" neutrophils "to" platelets. Although the authors rule out the contribution of platelet-derived EVs transferred to neutrophils (novel Figure 5I) in a test tube in their experimental conditions, they should at least acknowledge that it might still be occurring in vivo. Additionally, the literature on EVs and inflammation (extensive literature for this), neutrophil-derived EVs (e.g. Timar CI et al. Blood 2013 Jan 17;121(3):510-8; Lorincz AM et al. J. Leukoc. Biol. 2015 Oct;98(4):583-9 205) and EVs transfer from platelets to neutrophils (e.g. Dushez et al. PNAS 2015 Jul 7;112(27):E3564-73) is completely overlooked, which makes the reading often irritating. In agreement with this, on page 16, line 20, the authors must specify that what's new to their knowledge, is the shuttling from neutrophils to platelets (and not between neutrophils and platelets). The authors should recognize the extensive work in the literature concerning EVs and inflammation, EV transfer from other cells, notably platelets, to neutrophils, and emphasis must be given to their key findings, which is the transfer of EV "from" neutrophils "to" platelets, which is totally novel and exciting.

2. Another clarification is needed regarding the mechanism implicating GPIb and Mac1. In Figure 4 F-N, the authors demonstrate that EVs can rescue GPIb blockade, consistent with their observations that GPIb is implicated in EV release (Fig 4A). This would also mean that GPIb -Mac-1 interaction is subsequently not required. Yet, in Figure 5, Mac-1 blockade inhibits the transfer of EVs to neutrophils, and they suggest that Mac-1 is necessary for uptake. As Mac-1 is thought to be the counter-receptor for GPIb, and that the authors have ruled out Mac-1-Fibrinogen-IbIIIa interactions, how do the authors reconcile these contradictory observations?

Minor:

Page 6 line 16; "productice"

Reference is missing page 8 line 24.

Page 9, line 17; one word is missing after "impaired..."

Reference page 17 (Sreemkumar 2014) needs to be formatted.

Reply to reviewers' comments

Reviewer 1

This is a very comprehensive study that elucidates mechanism of transcellular synthesis via exosomes between platelets and neutrophils. I have three major issues that need to be resolved with additional experiments.

Response: We thank the reviewer for the positive comments.

1) How is P-selectin upregulated on platelets in vivo in infections? Please demonstrate that this actually occurs. Figure 1 shows that the effects of fMLP and ADP are blocked by P-selectin antibody in a test tube but these effects may not be occurring in vivo. Why did the authors not add E.coli rather than ADP and fMLP? One clear issue is that while ADP causes P-selectin expression E.coli and/or LPS do not activate P-selectin very well. This reviewer is not doubting the data, simply the disconnect between in vivo and in vitro experiments.

Response: We agree with the reviewer. It has been shown that direct platelet stimulation with LPS or *E. coli* does not lead to substantial upregulation of platelet P-selectin surface expression *in vitro* (see Andonegui, Blood 2005;106(7):2417-23). However, bacterial infections are known to cause a local and systemic inflammatory response including the release of formylated peptides and pro-inflammatory mediators causing platelet activation *in vivo*. Thus, we have chosen fMLP and ADP as *in vitro* agonists. To directly demonstrate upregulation of P-selectin surface expression on platelets *in vivo*, we induced pneumonia by *E. coli* instillation in WT mice *in vivo*, isolated circulating platelets from blood 4, 12 and 24 hours later and analyzed platelet P-selectin expression by flow cytometry. We demonstrated that platelets from *E. coli* treated mice significantly expressed P-selectin on their cell surface compared to platelets isolated from saline-treated mice.

The following paragraph was added to the results section (page 10, line 1):

“Bacterial infections are known to cause a local and systemic inflammatory response including the release of formylated peptides and pro-inflammatory mediators causing platelet activation *in vivo*. To directly demonstrate upregulation of P-selectin surface expression on platelets *in vivo*, we induced pneumonia by *E. coli* instillation in WT mice, isolated circulating platelets from blood after 4, 12 and 24 hours and analyzed platelet P-selectin expression by flow cytometry. We demonstrated that platelets from *E. coli* treated mice expressed significantly more P-selectin on their cell surface compared to platelets isolated from saline-treated mice (Supplemental Figure 5e).”

2) Blocking P-selectin in vivo reduced the number of neutrophils recruited into BAL but this does not necessarily invoke a role for platelets. Bone marrow chimeras could delineate between platelets and endothelium.

Response: We agree with the reviewer and performed additional experiments with bone marrow chimeric mice.

The following paragraph was added to the results section (page 9, line 20):

“To specifically demonstrate that platelet and not endothelial P-selectin is responsible for these effect we transplanted bone marrow from P-selectin deficient donor mice (*Selp*^{-/-}) into lethally irradiated WT recipient mice and vice versa. Pneumonia was induced in these mice 6 weeks after bone marrow transplantation. Mice transplanted with *Selp*^{-/-} bone marrow showed a significantly reduced number of neutrophils in the BAL (Supplemental Figure 5a) and increased number of CFUs in the BAL, lung and spleen after *E. coli* instillation (Supplemental Figure 5b-d) compared to control mice transplanted with WT bone marrow.”

3) It is strange that busulfan depletion of platelets prevented neutrophils from entering BAL. The authors need to provide some evidence that the busulfan does not affect neutrophil biology. In fact other approaches to depleting platelets would be important.

Response: We agree with the reviewer and now demonstrate that ICAM-1 binding, phagocytosis and transmigration are not altered in neutrophils isolated from busulfan-treated mice. Additionally, we depleted platelets in WT mice using an anti-platelet antibody (clone R300, Emfret Analytics) and induced a pneumonia. These experiments demonstrate the same results as platelet depletion with busulfan.

The following paragraph was added to the results section (page 6, line 21):

“Platelet depletion by busulfan decreased the number of neutrophils in the BAL (Figure 2B) and significantly increased bacterial burden in the BAL, lung and spleen (Figure 2c-e) after *E. coli* instillation compared to vehicle-treated mice. The same effect was observed when platelets were depleted with a different approach utilizing a platelet-depleting antibody (Supplemental Figure 2a-d).”

The following paragraph was added to the results section (page 7, line 3):

“In order to exclude adverse effects of busulfan-treatment on neutrophil function, we isolated neutrophils from control- and busulfan-treated mice and analyzed ICAM-1 binding, transmigration and phagocytosis of *E.coli* particles *in vitro*. These experiments showed that all these neutrophil functions were not significantly altered by busulfan (Supplemental Figure 2e-g).”

Furthermore, we have shown before (Rossaint, Blood. 2014;123(16):2573-84; Zarbock JCI. 2006;116(12):3211-9) that busulfan-induce platelet depletion does not significantly alter differential white blood cell count at the day of experiment 19 days after initial busulfan treatment.

Reviewer 2

The manuscript explores an interesting topic and the proposed exosome signaling mechanism is novel enough. The requirement of neutrophils and platelets in vivo shown in Figures 2 and 3 are convincing. However there are several major technical issues and over interpretations in experiments assessing the exosome mechanism. The authors make a strong case that platelets are important for neutrophil recruitment and that the TxB₂ pathway is important in vivo. They also make a good case for neutrophils regulating the process. But the involvement of exosomes needs to be better analyzed and substantiated. Please see specific suggestions for Figures 1 and 4.

Response: We thank the reviewer for the positive comments.

While most of the manuscript examines neutrophil recruitment in a convincing fashion, the NET data are weak and not convincing. NETs are not even mentioned in the abstract. It might be better to remove these data from the manuscript as it offers little and weakens the overall quality.

Response: We agree with the reviewer and removed the NET data from the manuscript.

Finally the description of the experiments in the main text and legends is poor and in many cases (example figure 4) it is unclear how the experiments were performed. Also, the figure letters are lower case in the figures and capital in the text.

Response: We agree with the reviewer and improved the description of the experiments in the main text and the figure legends. Changes are highlighted in the revised manuscript. We also corrected the figure labeling according to the journal's format.

Specific comments:

Figure 1b is misinterpreted. When platelets are loaded with arachidonic acid they produce significant amounts of labeled TxB₂. There is a two fold increase in radio labeled TxB₂ which could be explained by having twice as many cellular sources. The data suggest that neutrophils might be able to supply arachidonic acid, but not "that neutrophils provide the majority of arachidonic acid required for maximal TxB₂ production" (pg. 6). At this stage all we know from fig 1a is that both cells are needed and from fig. 1b that both contribute. Perhaps comparing to the cells alone might help show that the neutrophils can produce TxB₂ themselves which will then hint that they have to donate arachidonic acid to platelets. Perhaps reordering the experiments might get around this problem.

Response: We agree with the reviewer and rephrased the mentioned part of the results section to emphasize that indeed both cell types are needed for maximum TxB₂ production and that both cell types contribute arachidonic acid to this process.

The following paragraph was added to the results section (page 5, line 15):

“This assay revealed that neutrophils provide a significant amount of arachidonic acid and that maximum TxB₂ production is only achieved when both cellular sourced of arachidonic acid (i.e. neutrophils and platelets) cooperate in prostaglandin production (Figure 1b).”

Figure 1c and d. The labels are so small that they are impossible to read. Also, it is unclear which cells were labeled in these experiments. This is crucial because the data show that Cox enzymes are not important in neutrophils but only in platelets. Therefore, one can't use the mutants to decipher where the arachidonic acid is coming from as it would be the case if it were processed in neutrophils. The amount of radioactive signal based on Fig. 1b suggests that both cells are labeled here. Given the results here, it should only be neutrophils labeled in order to assess their transfer of arachidonic acid.

Response: We agree with the reviewer and improved the visualization of Figure 1c and d for better readability. In these experiments, only neutrophils were labeled with C¹⁴-AA. We clarified this in the manuscript.

The following paragraph was added to the results section (page 5, line 21):

“To differentiate the role of Cox1 and Cox2 in platelets and neutrophils in TxB₂-C¹⁴ production using arachidonic acid from neutrophils, we isolated neutrophils and platelets from WT, *Cox1*^{-/-} and *Cox2*^{-/-} mice and incubated only the neutrophils with AA-C¹⁴. Afterwards, neutrophils and platelets were coincubated and TxB₂-C¹⁴ production and total TxB₂ levels were analyzed. We observed that only Cox1 in platelets is necessary for TxB₂ production utilizing arachidonic acid from neutrophils (Figure 1c-d).”

Figure 2i. It is not clear where the platelets are. Immunofluorescence microscopy is required.

Response: To further visualize the allocation of platelets and neutrophils during *E. coli* induced pneumonia we stained neutrophils, platelets and PECAM as an endothelial marker of the pulmonary microvasculature and analyzed lung sections from *E. coli* treated animals by confocal microscopy. These experiments confirm that the interaction of neutrophil and platelets indeed appears in the intravascular space during *E. coli* induced pneumonia. This is in line with the EM images in Figure 2i, where the interaction of neutrophils and platelets occurs within a pulmonary capillary in the vicinity of 2 erythrocytes present in that vessel. For better interpretability of the EM image in Figure 2i we improved the labeling and now additionally point out the erythrocytes and the capillary wall in the figure.

The following paragraph was added to the results section (page 7, line 18):

“To further visualize the allocation of platelets and neutrophils during *E. coli* induced pneumonia we stained neutrophils, platelets and PECAM as an endothelial marker of the pulmonary microvasculature and analyzed lung sections from *E. coli* treated animals by confocal microscopy (Figure 2j).”

The following paragraph was modified in the figure legend (page 33, line 4):

“Ultrathin cross-sectioned lung tissue imaged by transmission electron microscopy from lung tissue of WT after inducing pneumonia showing a neutrophil (asterisk) in close proximity to a platelet (#) and 2 erythrocytes (§) within the boundaries of the capillary wall (arrow).”

Figure 4c. The authors need to substantiate the microscopy with biochemistry to show that exosomes are taken up by platelets. They can incubate platelets with exosome preparations from unstimulated and stimulated neutrophils, isolate away the platelets and show the incorporation of Mac-1 or other neutrophil exosome markers into platelets by WB.

Response: We agree with the reviewer and thank him/her for this helpful suggestion. We have performed the assay suggested by the reviewer.

The following paragraph was added to the results section (page 12, line 20):

“To further prove that EVs are internalized by platelets, isolated neutrophil-derived EVs were co-incubated with platelets, followed by isolation of washed platelets. Western blot analysis of these platelet preparations demonstrated the presence of Mac-1 in platelet lysates after co-incubation with neutrophil-derived EVs (Figure 5g).”

Figure 4e-h. The authors use neutrophils here. However they should repeat these functional experiments with isolated exosomes to prove that they are sufficient to mediate the effect of neutrophils.

Response: We agree with the reviewer performed the suggested experiments.

The following paragraph was added to the results section (page 11, line 3):

“Having shown that the blocking GPIIb/IIIa reduces EV release from neutrophils, we investigated thromboxane production and observed that GPIIb/IIIa blockade reduces both the TxB₂-C¹⁴ release, indicating reduced usage of neutrophil-derived arachidonic acid by platelets, as well as total thromboxane production (Figure 4f-g). In agreement with this finding, GPIIb/IIIa blockade (clone Xia.B2, 5µg/ml) reduced platelet Cox1 activity (Figure 4h). The addition of isolated EVs to the platelet-neutrophil coculture after GPIIb/IIIa blockade was able to rescue TxB₂-C¹⁴ release, total thromboxane production and platelet Cox1 activity (Figure 4f-h).”

Supplementary figure 4c. These data are problematic. First, the legend states that red is staining for histone H3. This antibody should stain all nuclei then, not only NETs. But it does not. Why? The authors should use an antibody against citrullinated histone H3 instead. Furthermore, MPO is confusingly shown in yellow (!) in a merged image where green and red are present which makes it impossible to judge which marker is which. Overall it is impossible to deduce anything about NETs from these images other than there are fewer neutrophils present in Cox 1 -/- mice which is of course not novel. The MPO/DNA complex ELISA alone is not sufficient to make the case for NETosis because secreted MPO could associate with bacterial or host DNA as well. Moreover, the effects based on these assays are not impressive.

Response: As suggested by the reviewer we removed the NET data from the manuscript.

Supplementary Figure 4e. The WT sample shows on structure that is stains for H3 (not citrullinated H3) but no other nucleus stains for H3. This is strange. Furthermore, the so called "NET" seems to lack DNA staining. Finally, only one potential "NET" is shown in an image of several intact neutrophils. The main text and legend do not specify how these neutrophils were activated. Only that they were incubated with activated platelets. According to Paul Kubes LPS activation of platelets to make NETs works in very few cells and only under flow conditions. Therefore, robust quantitation is needed to examine the effects of Cox deficiency in a system where the positive control is very poor to begin with. Furthermore, if LPS was used, it is odd that neutrophils do not appear polarised and activated.

Response: As suggested by the reviewer we removed the NET data from the manuscript.

Reviewer 3

The work by Rossaint and colleagues relates to two interesting aspects; 1) interplay between platelets and neutrophils in the combat of bacterial infections in lungs, and 2) transcellular metabolism of arachidonic acid (AA) involving platelets and neutrophils during bacterial infection. For their study, they used two models of lung infections. Although several studies have reported evidence of collaborations between neutrophils and platelets for the generation of eicosanoids during infectious processes, one interesting aspect suggested in this present study is the involvement of neutrophil-derived exosomes. According to the authors, neutrophils release exosomes, which AA cargo is used by platelet machinery to synthesize thromboxane. Based on their observations, interactions between platelets and neutrophils implicate P-selectin and PSGL-1. Furthermore, thromboxane induces endothelial cell expression of ICAM-1, thus permitting intravascular crawling and neutrophil recruitment at the site of bacterial infection. Ablation of this pathway (using KO mice, inhibitors, and blocking abs) leads to aggravated infection and reduction of mouse survival. It is a very interesting study, containing a series of convincing and well-designed experiments.

Response: We thank the reviewer for the positive comments.

Some concerns require clarification:

Interactions between neutrophils and platelets, via P-selectin and PSGL-1 is not novel; it has been described by several groups, and some of the key studies have been cited by Rossaint et al. What is important in this study, however, is the transcellular metabolism of AA mediating thromboxane A2 production. Transcellular metabolism of AA between platelets and other cellular lineages, including neutrophils, has been known for a long time (see for instance the work from Serhan C., Collier B., FitzGerald G., Boyce JA., and the more recent study from Levy BD in PNAS 2014 Nov 18;111(46):16526-31). The authors claim that the mechanism, revealed in their study, is novel and significant because it implicates neutrophil-derived exosomes. To prove this, they labeled AA with C14 in neutrophils or in platelets, and they could determine that most of the thromboxane produced transcellularly implicated AA from neutrophils and cox-1 from platelets.

Response: We thank the reviewer for the positive comments.

1) One major issue here is that there is no experiment that excludes the (most plausible) possibility that platelet-derived cox-1 is actually shuttled to neutrophils via extracellular vesicles (e.g. microparticles), where it metabolizes neutrophil-derived AA. Such pathway has been reported in several studies (for instance, see Barry OP, JCI 1997 May 1;99(9):2118-27, and the more recent Duchez AC, PNAS 2015 Jul 7;112(27):E3564-73), which demonstrate shuttling of cox-1 between platelets and neutrophils. Hence, platelets and/or platelet microparticles seem to be internalized in neutrophils in Figure 4D,E.

Response: We agree with the reviewer that this is an important issue and analyzed a possible contribution of Cox1 shuttling from platelets to neutrophils. To investigate this, we co-incubated fMLP-activated neutrophils and ADP-activated platelets (ratio 1:10), isolated the neutrophils by density gradient centrifugation and analyzed Cox-1 activity in these neutrophils *in vitro*. The same number of neutrophils was used each time we compared the Cox-1 activity in this experiment. However, we did not detect a significant impact on Cox1 activity in neutrophils after co-incubation with platelets. Although of course this finding

does not rule out a minor contribution of platelet-derived Cox1 in neutrophils, at the very least it rules out that this inverse shuttling mechanism dominates over the one we show here.

The following paragraph of the results section was rephrased accordingly (page 13, line 9):

“Previous reports on reverse transcellular communication by EV transport from activated platelets to neutrophils imply a possible Cox1 shuttling from platelets to neutrophils. However, we investigated the activity of Cox1 after co-incubation of isolated WT neutrophils with either WT or *Cox1*^{-/-} platelets and could not detect a significant difference in neutrophils (Figure 5i).”

2) A second issue regards the notion of intercellular communication via exosomes. The authors claim that exosomes, very specifically, are implicated in the transcellular metabolism of AA; however there is no experiment that confirms the involvement of exosomes, and rule out the implication of microvesicles or other types of extracellular vesicles. In fact, the marked differences between extracellular vesicles produced by cytoplasmic membrane budding and those stored in multivesicular bodies or apoptotic bodies were totally overlooked. Hence, the authors mistakenly state that "exosomes are also termed microvesicles or microparticles", which is absolutely incorrect (see introduction). Furthermore, although the authors used commercial kits to "isolate" and "enumerate" exosomes, it is well known that other particles (micro-particles/microvesicles, ectosomes, apoptotic bodies, protein aggregates, secretory vesicles, etc.) can interfere in these kits and are in fact isolated together with exosomes, especially if no other gradient/velocity centrifugations are performed. The experiments in Figure 5 do not permit to confirm that the vesicles implicated are exosomes. The authors should refer to most up to date notions on extracellular vesicles and should follow recommendations from the international society of extracellular vesicles and the ISTH in their definition of exosomes and the design of their experiments (see Lotvall, JEV 2014 Dec 22;3:26913). They have to correctly define what material they are studying, and provide rationale for their definition.

Response: We apologize for our “loose” terminology as we are not specialists in this field. We therefore agree with the reviewer in the necessity to further characterize the nature of the extracellular vesicles reported in this study. As suggested by the reviewer, we further analyzed our study material according to the guidelines published by the International Society of Extracellular Vesicles as published (Lötvall et al., JEV 2014). As we agree with the reviewer that we cannot claim that these are only exosomes (as seems to be the case with the majority of available isolation protocols), we changed the denomination of our isolated particles to “extracellular vesicles (EV)” throughout this study.

The following paragraph was added to the Supplemental Materials (page 2, line 6).

“To further characterize the investigated extracellular vesicles, we analyzed our study material according to the guidelines published by the International Society of Extracellular Vesicles³. As the concomitant contamination of extracellular vesicle preparations with apoptotic cell bodies may obscure the studied effects of extracellular vesicles, we analyzed stained EV supernatants with Trypan Blue staining and could not detect significant amounts of contamination with apoptotic cell remnants in our EV preparation. As suggested by the International Society of Extracellular Vesicles we also investigated the expression of proteins that neutrophil-derived EVs may inherit from their origin cell (i.e. neutrophils). Western blot analysis revealed that the integrins Mac1, the integrin subunits CD11a (α_L) and CD18 (β_2) as well as PSGL-1 (CD162) are present in lysates from neutrophil-derived EV preparations, all of which are also expressed on the membrane of neutrophils, and the commonly EV-associated molecule CD63 (Supplemental Figure 6a). In contrast, the histones H2A and H3 could not be detected in lysates from neutrophil-derived EV preparations, as these proteins are usually nucleus-associated and do not associate with endosomal structures. To also show the functional dose-response relationship of the neutrophil-derived EV

isolates we blocked endogenous EV liberation in a stimulated platelet-neutrophil co-culture in vitro and substituted isolated EVs at different concentrations to measure the amount of TxB2 in this system and could observed increasing TxB2 generation with increasing substitution amounts of isolated neutrophil-derived EVs (Supplemental Figure 6b).”

3) When "exosomes" are utilized in their in vitro and in vivo studies, there is no indication of the concentrations used. Without this information, it is impossible to the reviewer to assess of the relevance of the experiments. Furthermore, extracellular vesicles in BAL during infection could be determined using appropriate approaches.

Response: We provide relevant information on EV concentrations in the methods section of our revised manuscript (Supplemental Material: page 2, line 5). We also analyzed EV levels in the BAL fluid from mice after inducing *E. coli* induced pneumonia.

The following paragraph was added to the results section (page 13, line 6):

“The induction of an *E. coli*-induced pneumonia caused a modest increase in the concentration of EVs in the BAL. However, no significant decrease in the concentration of EVs where detectable in platelet-depleted mice after *E. coli* challenge (data not shown).”

The following paragraph was added to the discussion (page 16, line 20):

“Interestingly, we did not detect significant alterations in the concentration of EVs after *E. coli* pneumonia induction between control and platelet-depleted mice. This may be explained by the fact that EV shuttling from neutrophils to platelets represents a process in the immune cell recruitment taking place at the emigration of neutrophils from the microvasculature into the lung tissue, eventually enabling the neutrophils to enter the alveolar space. Another explanation may be that different mechanisms of EV generation in the alveolar compartment and secretion into the BAL are involved apart from platelet-neutrophil interactions. The exact temporal-spatial contribution of EV shuttling to the distinct process of neutrophil recruitment has yet to be investigated in detail to gauge the role of detectable neutrophil-derived EVs in the bronchoalveolar lavage fluid.”

4) In Figure 1B, the authors mention that most of the thromboxane is synthesized from neutrophil-derived AA. However, this is based on the assumption that both platelets and neutrophils were equally labeled with C14-AA. Since important quantities of TxB2 also originate from platelet-derived AA, it is important to confirm this authors' statement.

Response: Arachidonic acid is unspecifically integrated into the cell membrane. Because platelets are considerably smaller than neutrophils, the amount of cell membrane per cell is also lower in a platelet compared to a neutrophil. For this reason, we have chosen to compare the radioactivity of C¹⁴-AA in cell lysates from platelets and neutrophils with equal cell masses. In order to ensure equal labeling of platelets and neutrophils in this assay, cells were washed and lysed after incubation with C¹⁴-AA and the radioactive counts per minute (cpm) of cell lysates with equal protein content (indicating equal cell masses) were analyzed using the β -radiation counter. These assays revealed equal loading of the labeled AA into both neutrophils and platelets.

The following paragraph was added to the results section (page 5, line 18):

“In order to ensure equal labeling of platelets and neutrophils in this assay, the radioactive counts per minute (cpm) of cell lysates with equal protein content (indicating equal cell masses) were analyzed using a β -counter to verify equal C¹⁴-AA loading.”

5) *In addition, it is surprising that platelets, which are recognized for their important pool of membrane, require AA supply from other cells. Thromboxane synthase is inactivated by its products (a process known as suicide inactivation), and substrate availability in platelets is generally not limiting. The authors should comment on this.*

Response: We agree with the reviewer and included this interesting point in the discussion (page 17, line 12):

“On first glance it may appear surprising that platelets depend on arachidonic acid from neutrophil-derived EVs to efficiently generate thromboxane, as they possess cellular arachidonic acid storages themselves. However, this phenomenon may be explained by our finding that the uptake of neutrophil-derived EVs not only provides substrates to platelets, but also increases Cox1 activity in platelets, thus optimizing and synchronizing the usage of substrates for thromboxane generation.”

6) *The authors rapidly rule out IIbIIIa in neutrophil-platelet interactions, using an in vitro system where fibrinogen is absent. Such assay should be performed in presence of plasma, and further confirmation in vivo will be needed to exclude a contribution of IIbIIIa.*

Response: We repeated the *in vitro* assays with tirofiban using platelets in plasma.

The following paragraph of the results section was rephrased accordingly (page 6, line 1):

“To identify molecular adhesion molecules required for intercellular interactions and exchange of arachidonic acid during thromboxane production by platelets, we used a blocking anti-P-selectin antibody (RB40.34) and anti-PSGL-1 antibody (4RA10) as well as tirofiban (antagonist of the platelet integrin $\alpha_{IIb}\beta_{III}$) and demonstrated that $\text{TxB}_2\text{-C}^{14}$ production and overall TxB_2 production was reduced after blockade of P-selectin or PSGL-1, but not after inhibiting the platelet integrin $\alpha_{IIb}\beta_{III}$ (Figure 1e-f). Notably, the activity of Cox1 was increased in the presence of neutrophils, and again this effect was reversed after blocking P-selectin or PSGL-1, but not $\alpha_{IIb}\beta_{III}$ (Figure 1g). Blocking the platelet integrin $\alpha_{IIb}\beta_{III}$ by tirofiban also did not show an effect in the same *in vitro* assays in the presence of fibrinogen in plasma (data not shown).”

However, reporting this data we only rule out a role of the platelet integrin $\alpha_{IIb}\beta_{III}$ in substrate exchange between neutrophils and platelets. We do not rule out a role for $\alpha_{IIb}\beta_{III}$ in formation of neutrophil extracellular traps (implicated in bacterial killing) and neutrophil recruitment to the lungs, as we have previously reported (Rossaint et al., Blood 2014;123(16):2573-84) in a murine model of ventilator-associated lung injury (VILI). We have also blocked the platelet integrin $\alpha_{IIb}\beta_{III}$ using tirofiban in the *E. coli* induced pneumonia in this study and observed decreased neutrophil recruitment and impaired bacterial killing.

The following paragraph was added to the results section (page 9, line 16):

“We previously reported impaired neutrophil recruitment to the lungs after blocking the platelet integrin $\alpha_{IIb}\beta_{III}$ using tirofiban in a murine model of ventilator-associated lung injury, and we could also observe reduced neutrophil recruitment and impaired bacterial clearance in the *E. coli* induce pneumonia model after tirofiban administration (Figure 3b-e).”

7) Although the assay that revealed transcellular metabolism of AA between platelets and neutrophils is consistent with *in vivo* data implicating thromboxane (using antagonist SQ29548), more information on the complete set of eicosanoids produced between neutrophils (exosomes) and platelets would be informative. For instance, what do we know on the levels of lipoxin A4, cystenyl leukotrienes in this system?

Response: We agree with the reviewer and analyzed the biosynthesis of LTB4 and LTC4 (for cystenyl leukotrienes) in our *in vitro* co-culture model with platelets and neutrophils. The biosynthesis of both leukotrienes is known to involve transcellular metabolism between platelets and leukocytes, with LTB4 synthesis by neutrophils relying on transcellular substrate transport from platelets to neutrophils and platelets requiring neutrophil-derived LTA4 for transformation to LTC4 (see Capra et al., *Biochim Biophys Acta*. 2015;1851(4):377-82). Interestingly, the concentration of both LTB4 and LTC4 were increased under stimulated compared to control conditions. However, blocking GPIIb/IIIa did not significantly impact LTB4 or LTC4 production, indicating the implication of different modulatory molecular mechanisms

The following paragraph was added to the results section (page 11, line 15):

“Besides thromboxane, the biosynthesis of leukotrienes is also known to involve platelet-neutrophil interaction. While LTB4 synthesis by neutrophils relies on transcellular substrate transport from platelets to neutrophils, platelets require neutrophil-derived LTA4 for transformation to LTC4 (see Capra et al., *Biochim Biophys Acta*. 2015;1851(4):377-82). The concentrations of both LTB4 and LTC4 were increased under stimulated compared to control conditions. However, blocking GPIIb/IIIa did not significantly impact LTB4 or LTC4 production, indicating the implication of different modulatory molecular mechanisms in comparison to thromboxane biosynthesis (Figure 4o-p).”

Minor

1) Supplemental Figure 2 comes before supplemental Figure 1

Response: We apologize for the mistake and corrected the figure ordering accordingly.

2) It should be indicated when BAL and blood were collected after bacteria instillation. In conditions where mouse survival was impaired, were the fluids collected prior or after death? Were all the mice used for the generation of the data (BAL, CFU, TxB2 etc.), or only those that (did not) survived?

Response: For clarification, we included the following paragraph in the methods section (page 20, line 12):

“We used two different sets of experiments. For the first set of experiments, animals were challenged with 6×10^6 viable *E. coli* per mouse. At this inoculation dose, all mice survived the 24 hours observation period. In a separate set of experiments, mice were challenged with a higher inoculation dose (8×10^6 viable *E. coli* per mouse) which allowed the survival analysis. After 24 hours, the mice challenge with 6×10^6 viable *E. coli* were sacrificed and the lungs were lavaged 5 times with 0.7 ml physiologic saline solution. The number of neutrophils in the BAL was counted using kimura staining.”

3) Some graphs are hard to read. See Figures 1C and D for instance.

Response: We agree with the reviewer and improved the visualization of Figure 1c and d for better readability.

4) *In the result section (Figure 5): "vehicle exocytosis" or vesicle exocytosis?*

Response: We changed “vehicle exocytosis” to “vesicle exocytosis”.

5) *Details on methods are frequently lacking. For instance, neutrophils in BAL were quantified using kimura staining, but there is no indication on the number of fields and cells counted. More information on this approach would be needed.*

Response: We agree with the reviewer and added the following paragraph to the method section (page 20, line 18):

“The number of neutrophils in the BAL was counted using kimura staining. Neutrophils were counted using an improved Neubauer counting chamber and an inverted cell culture microscope (Primovert, Carl Zeiss, Göttingen, Germany) equipped with a 10x0.75NA objective. A total of 4 fields with 16 standardized subfields in each individual field were counted for each sample.”

Reply to reviewers' comments

Reviewer 1

The authors did a great job of addressing my concerns this can be accepted.

Response: We thank the reviewer for the positive comments.

Reviewer 2

The authors have fully addressed all of my comments.

In my view the story is now very clear, experimentally sound well rounded.

Response: We thank the reviewer for the positive comments.

Reviewer 3

This is an improved version of their manuscript, with several additional experiments and clarifications. There are lingering concerns, however, that still need to be addressed.

1. One key concept that is brought by the authors is the directed EV shuttling "from" neutrophils "to" platelets. Although the authors rule out the contribution of platelet-derived EVs transferred to neutrophils (novel Figure 5I) in a test tube in their experimental conditions, they should at least acknowledge that it might still be occurring in vivo. Additionally, the literature on EVs and inflammation (extensive literature for this), neutrophil-derived EVs (e.g. Timar Cl et al. Blood 2013 Jan 17;121(3):510-8; Lorincz AM et al. J. Leukoc. Biol. 2015 Oct;98(4):583-9 205) and EVs transfer from platelets to neutrophils (e.g. Dushez et al. PNAS 2015 Jul 7;112(27):E3564-73) is completely overlooked, which makes the reading often irritating. In agreement with this, on page 16, line 20, the authors must specify that what's new to their knowledge, is the shuttling from neutrophils to platelets (and not between neutrophils and platelets). The authors should recognize the extensive work in the literature concerning EVs and inflammation, EV transfer from other cells, notably platelets, to neutrophils, and emphasis must be given to their key findings, which is the transfer of EV "from" neutrophils "to" platelets, which is totally novel and exciting.

Response: We agree with the reviewer and added the following paragraph to the discussion section (page 16, line 18):

“This is, to our knowledge, the first report indicating that intercellular transport of metabolic substrates is specifically mediated by EV shuttling between neutrophils and platelets, and not the other way from platelets to neutrophils. While we could identify this route of intracellular substrate transport from neutrophils to platelets *in vitro*, we cannot fully rule out that the vice versa route from platelets to neutrophils may also play a role *in vivo*. Indeed, it has been reported that platelet microparticles may be internalized into neutrophils in a 12(S)-HETE-dependent manner during inflammatory arthritis (Duchez, PNAS 2015). Previous studies also indicate that opsonized bacterial particles may induce the release of neutrophil-derived EVs with antibacterial properties (Timar, Blood 2013). Furthermore, different stimuli may induce the generation and release of different EV subsets with divergent molecular composition and antibacterial functions (Lorincz, JLB 2015).”

2. Another clarification is needed regarding the mechanism implicating GPIIb and Mac1. In Figure 4 F-N, the authors demonstrate that EVs can rescue GPIIb blockade, consistent with their observations that GPIIb is implicated in EV release (Fig 4A). This would also mean that GPIIb-Mac-1 interaction is subsequently not required. Yet, in Figure 5, Mac-1 blockade inhibits the transfer of EVs to neutrophils, and they suggest that Mac-1 is necessary for uptake. As Mac-1 is thought to be the counter-receptor for GPIIb, and that the authors have ruled out Mac-1-Fibrinogen-IIbIIIa interactions, how do the authors reconcile these contradictory observations?

Response: We agree with the reviewer and performed additional pneumonia experiments to show that GPIIb α blockade of isolated EVs *in vitro* before re-injection into GPIIb α -treated animals indeed prevents the rescue of the phenotype and leads to impaired neutrophil recruitment and increased bacterial CFU numbers in the BAL, lung and spleen of these animals compared to GPIIb α -treated animals which received untreated, isolated EVs. Thus, we assume that the amount of GPIIb α antibody injected into the recipient animals to prevent endogenous EV generation did not suffice to also completely block the internalization of the isolated EVs which were re-injected into these animals.

The following paragraph was added to the results section (page 13, line 13):

“Interestingly, the reconstitution of GPIIb α -treated animals with isolated EVs restored neutrophil recruitment and bacterial clearance after induction of pneumonia (Figure 4). We induced pneumonia in GPIIb α -treated animals that were consecutively reconstituted with isolated EVs that had been treated with the GPIIb α blocking antibody *in vitro* before re-injection into the animals. The direct GPIIb α blockade prevented the rescue of the phenotype and led to impaired neutrophil recruitment and increased bacterial CFU numbers in the BAL, lung and spleen of these animals compared to GPIIb α -treated animals which received untreated, isolated EVs (Supplemental Figure 6a-d). Thus, we assume that the amount of GPIIb α antibody injected into the recipient animals to prevent endogenous EV generation did not suffice to also completely block the internalization of the isolated EVs which were re-injected into these animals.”

Minor:

Page 6 line 16; "productice"

Response: We apologize for the mistake and changed the word to “productive”.

Reference is missing page 8 line 24.

Response: We would like to refer to the findings in Figure 1 of the present manuscript at this point. To clarify this, we change the passage accordingly (page 8, line 24):

“These findings are consistent with our previous observation that platelet-derived Cox1, but not Cox2, is needed for the synthesis of TxA₂ upon contact with neutrophils (see Figure 1).”

Page 9, line 17; one word is missing after "impaired..."

Response: We apologize for the mistake and changed the sentence accordingly (page 9, line 17):

“We previously reported impaired neutrophil recruitment to the lungs after blocking the platelet integrin $\alpha_{IIb}\beta_{III}$ using tirofiban in a murine model of ventilator-associated lung injury, and we could also observe reduced neutrophil recruitment and impaired bacterial clearance in the *E. coli* induce pneumonia model after tirofiban administration (Figure 3b-e).”

Reference page 17 (Sreemkumar 2014) needs to be formatted.

Response: We formatted the reference according to the journals instructions.